# Dual functionalized brain-targeting nanoinhibitors restrain temozolomide-resistant glioma via attenuating EGFR and MET signaling pathways

Xiangqi Meng[1,4], Yu Zhao[2,4], Bo Han[1], Caijun Zha[3], Yangong Zhang[1], Ziwei Li[1], Pengfei Wu[1], Tengfei Qi[1], Chuanlu Jiang[1]*, Yang Liu[2]* & Jinquan Cai[1]*

Activation of receptor tyrosine kinase (RTK) protein is frequently observed in malignant progression of gliomas. In this study, the crosstalk activation of epidermal growth factor receptor (EGFR) and mesenchymal-epithelial transition factor (MET) signaling pathways is demonstrated to contribute to temozolomide (TMZ) resistance, resulting in an unfavorable prognosis for patients with glioblastoma. To simultaneously mitigate EGFR and MET activation, a dual functionalized brain-targeting nanoinhibitor, BIP-MPC-NP, is developed by conjugating Inherbin3 and cMBP on the surface of NHS-PEG$_8$-Mal modified MPC-nanoparticles. In the presence of BIP-MPC-NP, DNA damage repair is attenuated and TMZ sensitivity is enhanced via the down-regulation of E2F1 mediated by TTP in TMZ resistant glioma. In vivo magnetic resonance imaging (MRI) shows a significant repression in tumor growth and a prolonged survival of mice after injection of the BIP-MPC-NP and TMZ. These results demonstrate the promise of this nanoinhibitor as a feasible strategy overcoming TMZ resistance in glioma.

[1] Department of Neurosurgery, The Second Affiliated Hospital of Harbin Medical University, Neuroscience Institute, Heilongjiang Academy of Medical Sciences, 150086 Harbin, China. [2] State Key Laboratory of Medicinal Chemical Biology, Key Laboratory of Functional Polymer Materials of Ministry of Education, College of Chemistry, Nankai University, 300071 Tianjin, China. [3] Department of Laboratory Diagnosis, The Second Affiliated Hospital of Harbin Medical University, 150086 Harbin, China. [4] These authors contributed equally: Xiangqi Meng, Yu Zhao. *email: jcl6688@163.com; yliu@nankai.edu.cn; caijinquan@hrbmu.edu.cn

Glioma is a devastating brain tumor with poor prognosis and low median survival time[1]. Standard treatment includes radiation and chemotherapy with the DNA alkylating agent temozolomide (TMZ)[2]. However, a large percentage of tumors are resistant to TMZ-induced DNA damage due to elevated expression of the DNA damage repair proteins[3]. Although clinical practices and experimental evidences have obviously revealed that biologic targeting therapy could act as an alternate or adjuvant treatment option for patients with TMZ-resistant gliomas[4], it cannot achieve the full therapeutic potential for treatment because of the complicated crosstalk among multiple signaling pathways[5].

The coactivation of receptor tyrosine kinase (RTK) proteins, including insulin-like growth factor receptor (IGFR), hepatocyte growth factor receptor (HGFR; also known as MET), fibroblast growth factor receptor (FGFR), vascular endothelial growth factor receptor (VEGFR) and the EGFR family[6], acts as an important mechanism by which cancer cells develop acquired chemoresistance[7]. As RTK members, the epidermal growth factor receptor (EGFR), mesenchymal–epithelial transition factor (MET) and two major downstream pathways (Ras/MAPK/ERK and Ras/PI3K/AKT)[8] trigger uncontrolled cell growth, angiogenesis, and metastasis and chemoresistance[9–11]. Crosstalk between EGFR and MET contributes to the poor efficacy in the clinical treatment and malignant progresses such as alternative DNA damage repair[12–16]. It is likely that dual inhibition of EGFR and MET is required to prevent the crosstalk due to compensation signaling through alternative RTK pathways and suppress the chemoresistance[17–19]. E2F1 pronounces resistance toward chemotherapy and poor patient prognosis[20], and is responsible for the transcriptional regulation of many important modules[21,22]. Therefore, it is critical to explore a potential effective targeting therapeutic strategy simultaneously targeting EGFR and MET to overcome TMZ resistance via E2F1 in glioma.

Compared with therapeutic antibodies, peptides have the advantage to more easily diffuse into tissue, yet they are prone to degradation and rapid clearance[23] with smaller molecular weight and less immune response after repeated administration[24]. Inherbin3 has exhibited the reduction of the EGFR signaling in tumor cells and the suppression of tumor growth in vivo via targeting EGFR and inhibits the EGFR phosphorylation[25,26]. cMBP, a MET targeting peptide which has been developed for tumor imaging and gene delivery[27], has a therapeutic efficacy to block MET signaling and provides a potential approach for the treatment of glioma[28]. However, glioma patients hardly benefit from targeting therapy because of the presence of blood–brain–barrier (BBB), a tightly sealed filter formed by blood vessels that prevents unwanted biomolecules and drugs from accessing the brain tissue[29,30]. Encapsulation of poly-2-methacryloyloxyethyl phosphorylcholine (PMPC), a zwitterionic polymer plays an important role in delivering small molecular drugs, antibodies or peptides to the central nervous system owing to its excellent biocompatibilities[31], empowers the nanocarriers with a biomembrane-like structure, providing a biocompatible chemical structure which is the basis for an ability to circulate for a long time in the blood. Therefore, PMPC-coating can improve therapy efficiency of biomolecules or drugs by assistance on excellent BBB penetration[32].

In this study, we develop a nanoinhibitor, BIP-MPC-NP, which can simultaneously mitigate the crosstalk between EGFR and MET signaling pathways contributing to the TMZ resistance by conjugating Inherbin3 and cMBP on the surface of NHS-PEG$_8$-Mal modified MPC-nanoparticle in glioma. In vitro and in vivo results show that BIP-MPC-NP downregulates E2F1 via ARE motifs with tristetraprolin (TTP) mediated by phosphorylated p38 and significantly reduces its

reduction of transcriptional activity on DNA damage repair modules to enhance the TMZ therapy effect. These results advance our understanding of the RTK signaling pathway in glioblastoma (GBM) and demonstrate the promise of this nanoinhibitor as a feasible strategy overcoming TMZ resistance in glioma (Fig. 1a).

## Results

**Synthesis and characterization of BIP-MPC-NP.** To further investigate the genomic aberration involved in the biological process of glioma, we employed The Cancer Genome Atlas (TCGA) copy number variation (CNV) datasets of glioma and found that amplification events of chromosome 7 were the major alteration in glioma, and the amplification of EGFR and MET gene copy numbers were the top two RTK gene variations on chromosome 7 in both lower grade glioma (LGG) and GBM (Supplementary Fig. 1a, b, Supplementary Data 1). EGFR and MET amplification levels were simultaneously associated with a number of genomic variations (gain threshold = 0.1, $P < 0.05$, Student's $t$-test), differential gene expression (fold change >1.2, $P < 0.05$, Student's $t$-test) and protein profiling regulation (fold change >1.2, $P < 0.05$, Student's $t$-test), implicated in Oncogenes, Tumor Suppressors, Translocated Cancer Genes, Protein Kinases, Cell Differentiation Markers, Cytokines and Growth Factors, Homedomain Proteins and Transcription Factors, indicating that EGFR and MET amplifications were widely implicated in biological processes in GBM (Supplementary Fig. 1c). Low EGFR or MET copy numbers indicated a longer overall survival (OS) and a longer progression-free survival (PFS) compared with high EGFR or MET copy numbers (Supplementary Fig. 1d, OS: EGFR, $P < 0.05$, MET, $P < 0.05$, log-rank test; PFS: EGFR, $P < 0.05$, MET, $P < 0.05$, log-rank test).

According to our previous work[33,34], we established TMZ-resistant GBM cells named LN229R, U87MGR and HG9R through TMZ treatment on LN229, U87MG and patient-derived GBM cell HG9 (Supplementary Fig. 1e). Gene expression profiling was performed for the LN229R and the parental cells LN229 and RTK genes had higher expression levels in LN229R cells than in LN229 cells, especially for EGFR and MET (Supplementary Fig. 2a, Supplementary Data 2), consistent with our and other teams' work[11,33,35]. LN229R exhibited higher levels of EGFR and MET expression and phosphorylation in mouse orthotopic models bearing GBM xenografts (Supplementary Fig. 2b, c). With RNAi knocking down the expression of EGFR and MET in TMZ-resistant cells (Supplementary Fig. 3, Supplementary Table 1), simultaneous reduction of EGFR and MET could significantly downregulate the phosphorylation levels of the downstream proteins (p-AKT, p-p38, p-STAT3 and p-p65) compared with single knockdown of EGFR or MET (Supplementary Fig. 4), indicating that simultaneous inhibition of EGFR and MET activation could serve as an effective strategy for attenuating the intricate network of cross-signaling involving MET and EGFR.

Toward the development of an effective simultaneous inhibition strategy of EGFR and MET activation in TMZ-resistant glioma, we looked for a rational design of nanoinhibitor simultaneously targeting EGFR and MET, and developed a nanoinhibitor, BIP-MPC-NP, which can simultaneously mitigate EGFR and MET activation by conjugating Inherbin3[26] (denoted as EGFR-binding peptide, EBP) and cMBP[28,36] (denoted as MET-binding peptide, MBP) on the surface of NHS-PEG$_8$-Mal modified MPC-NPs (Fig. 1b). The transmission electron microscope (TEM) image of the nanoinhibitors showed a spherical shape with an average diameter of 23.65 (±2.23) nm, which was further confirmed with dynamic light scattering (DLS)

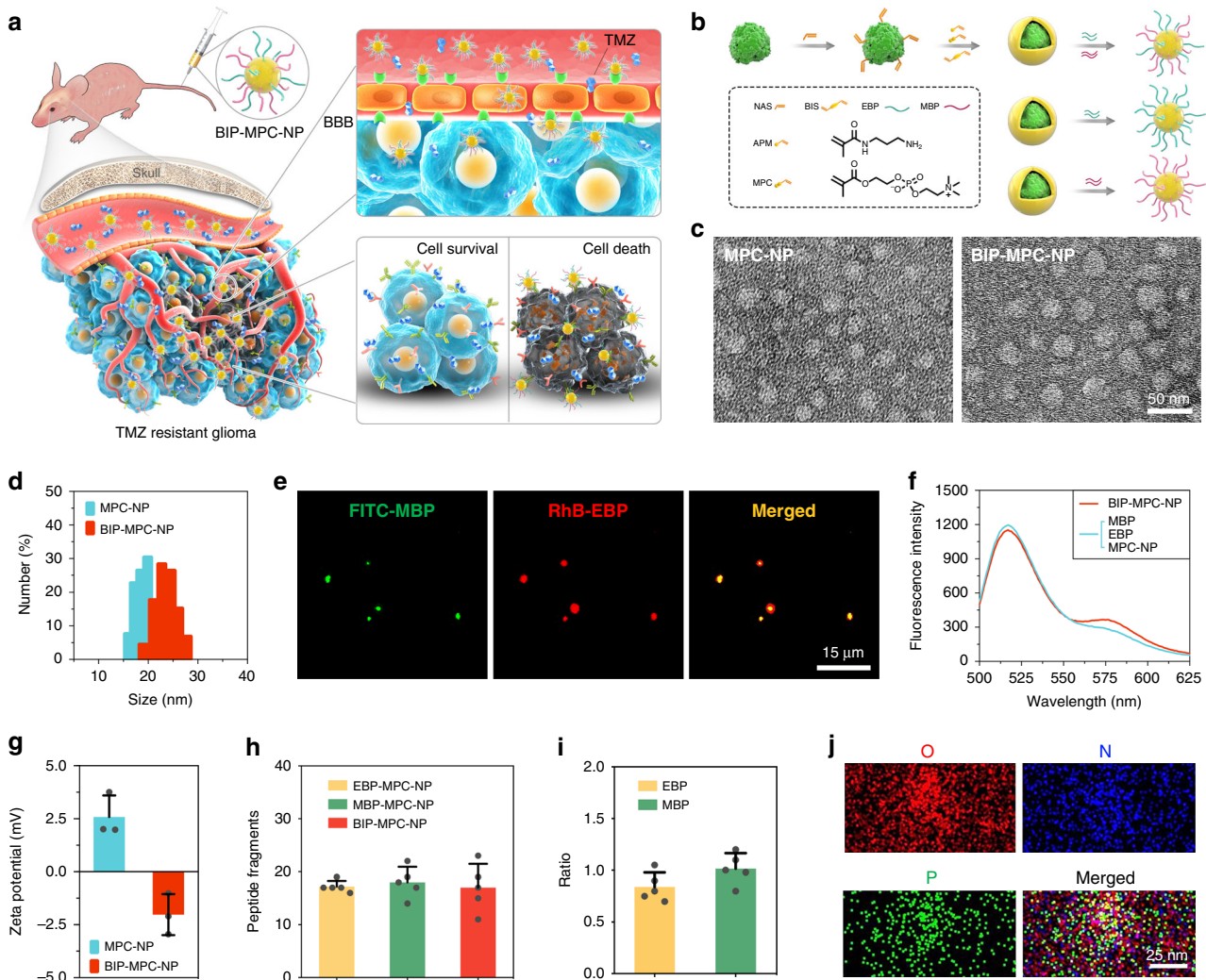

**Fig. 1 The synthesis and characteristics of nanoinhibitors. a** Schematic illustration and delivery process of the nanoinhibitor, BIP-MPC-NP, enhancing TMZ sensitivity against glioma via intravenous injection. **b** The schematic diagram revealing the establishment and structure of nanoinhibitors. **c** The transmission electron microscope (TEM) images of the nanoinhibitors. Scale bar = 50 nm. **d** The dynamic light scattering (DLS) measurements of BIP-NP and BIP-MPC-NP. **e** Fluorescence images presenting the co-localization of FITC-MBP and RhB-EBP for each BIP-MPC-nanoparticle. Scale bar = 15 μm. **f** Förster resonance energy transfer (FRET) analysis of BIP-MPC-NP and MBP/EBP/MPC-NP. **g** The zeta potential data of BIP-NP and BIP-MPC-NP ($n = 3$). **h** The quantitative analysis of peptide fragments per nanoinhibitor ($n = 5$). **i** The ratio of EBP and MBP on the nanoparticles ($n = 5$). **j** TEM micrograph exhibited the oxygen element (O), nitrogen element (N) and phosphorus element (P). Scale bar = 25 nm. The error bars in **g**, **h** and **i** represent the S.D. of three or five measurements.

measurements (Fig. 1c, d, Supplementary Fig. 5a). The DLS and polydispersity index (PDI) values of nanoinhibitors were also showed in Supplementary Table 2. To study the components of BIP-MPC-NP, EBP and MBP were labeled with fluorescein isothiocyanate (FITC-MBP) and rhodamine B isothiocyanate (RhB-EBP), respectively. Fluorescence images presented the co-localization of FITC-MBP and RhB-EBP for each BIP-MPC-NP (Fig. 1e). Förster resonance energy transfer (FRET) analysis confirmed an effective energy transfer from FITC-MBP to RhB-EBP, indicating that the EBP and MBP were closely associated within 10 nm (Fig. 1f). Compared to MPC-NP without peptides, the zeta potential of BIP-MPC-NP showed a significant decrease from 2.58 (±1.02) to −2.02 (±0.97) mV, confirming the conjugation of EBP and MBP on the surface of MPC-NP (Fig. 1g, Supplementary Fig. 5b). Further quantitative analysis suggested 15–20 peptide fragments per nanoinhibitor (Fig. 1h). The ratio of EBP and MBP on the nanoparticles was 0.83 (EBP/MBP) (Fig. 1i). Element mapping revealed that three typical elements in PMPC, including oxygen (O, red), nitrogen (N, blue) and

phosphorus (P, green), were observed in nanoinhibitors, indicating a successful integration of PMPC into the particles (Fig. 1j).

**The binding affinity and permeability of BIP-MPC-NP.** To elucidate the appropriate concentration of nanoinhibitors administered on cells, the IC50 assays were investigated after 24-h treatments in LN229R, and we chose 4 μM as the concentration for further experiments (Supplementary Fig. 6a). Then different nanoinhibitors were labeled with FITC (green) and in vitro fluorescence images showed the localization of EBP-MPC-NP, MBP-MPC-NP and BIP-MPC-NP on the surface of LN229R cells (red), while the MPC-NP without peptides showed negligible colocalization (Fig. 2a). Flow cytometric analysis showed that compared with cells treated with MPC-NP, those treated with EBP-MPC-NP, MBP-MPC-NP or BIP-MPC-NP had stronger fluorescent intensities (Fig. 2b), indicating that these nanoinhibitors could bind to the cell surface efficiently. With 5-h incubation, 5.2% of BIP-MPC-NP (4 μM) penetrated through the

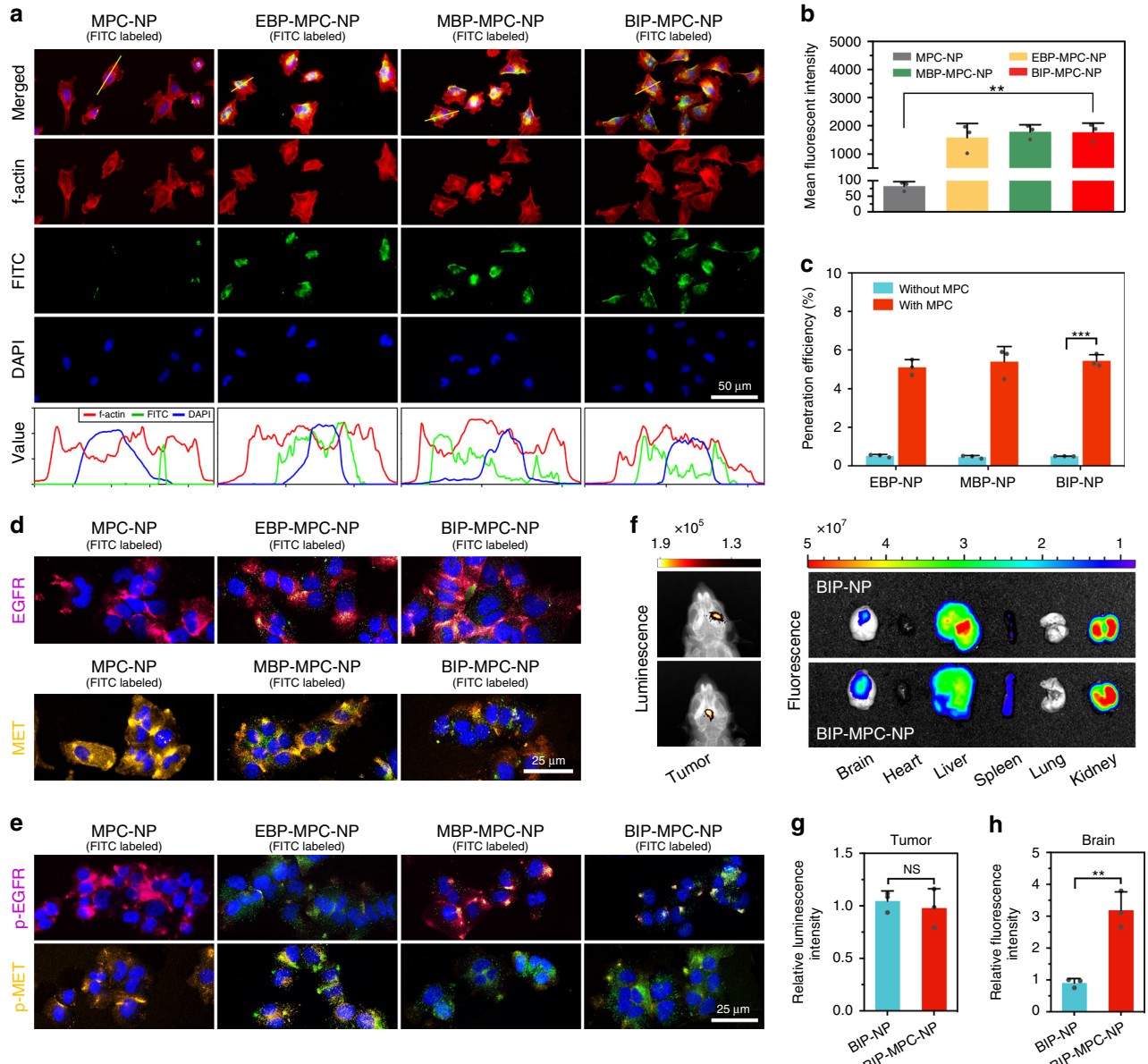

**Fig. 2 The biological activity of nanoinhibitors. a** The fluorescence assays showed the localization of EBP-MPC-NP, MBP-MPC-NP or BIP-MPC-NP on the LN229R cell surface. The gray value was displayed in line plots. Scale bar = 50 μm. **b** The flow cytometric analysis of cells with MPC-NP, EBP-MPC-NP, MBP-MPC-NP or BIP-MPC-NP in vitro ($n = 3$). **c** The transportation assays showed the permeability of EBP-MPC-NP, MBP-MPC-NP and BIP-MPC-NP in vitro ($n = 3$). **d** The binding affinity of MPC-NP, EBP-MPC-NP, MBP-MPC-NP or BIP-MPC-NP after penetrating BBB models. Scale bar = 25 μm. **e** The IF assays showed the attenuation of p-EGFR and p-MET by administration of MPC-NP, EBP-MPC-NP, MBP-MPC-NP or BIP-MPC-NP in BBB model. Scale bar = 25 μm. **f** Ex vivo fluorescence and bioluminescence images of the visceral organs after the injection of BIP-NP or BIP-MPC-NP in mice. **g, h** The histogram summarized the relative luminescence intensity and the relative fluorescence intensity of tumor-bearing brains. The error bars in **b**, **c**, **g** and **h** represent the S.D. of three measurements ($n = 3$). $P$ value is determined by Student's $t$-test. Significant results are presented as NS non-significant, **$P < 0.01$, or ***$P < 0.001$.

bEnd.3 layer (Fig. 2c, Supplementary Fig. 6b). After penetrating BBB model, the nanoinhibitors also had strong fluorescent intensities (Supplementary Fig. 6c) and the binding affinity targeting EGFR and MET (Fig. 2d). The expressions of p-EGFR and p-MET were simultaneously attenuated by BIP-MPC-NP penetrating BBB model (Fig. 2e).

To further evaluate the capability of the nanoinhibitors in crossing BBB in vivo, we employed LN229R tumor-bearing mice to perform this study. Cy5.5-labeled BIP-MPC-NP and BIP-NP were injected via the tail vein at a dosage of 100 μL (4 μM). The mice were then sacrificed, and the brains were harvested for

ex vivo imaging 5 h post the injection. According to the results (Fig. 2f–h, Supplementary Fig. 6d, e), the fluorescence signal of Cy5.5-labeled BIP-MPC-NP was observed clearly in the mouse brains and showed a 3.2-fold higher increase than BIP-NP, indicating effective BBB permeability. The ex vivo fluorescence images of the sliced tumor-bearing brain tissues showed the enhanced distribution of Cy5.5-labeled BIP-NP with MPC (Supplementary Fig. 6f). These results demonstrated that the nanoinhibitors were internalized into glioma tissues in vivo and BIP-MPC-NP had a more effective BBB permeability than BIP-NP.

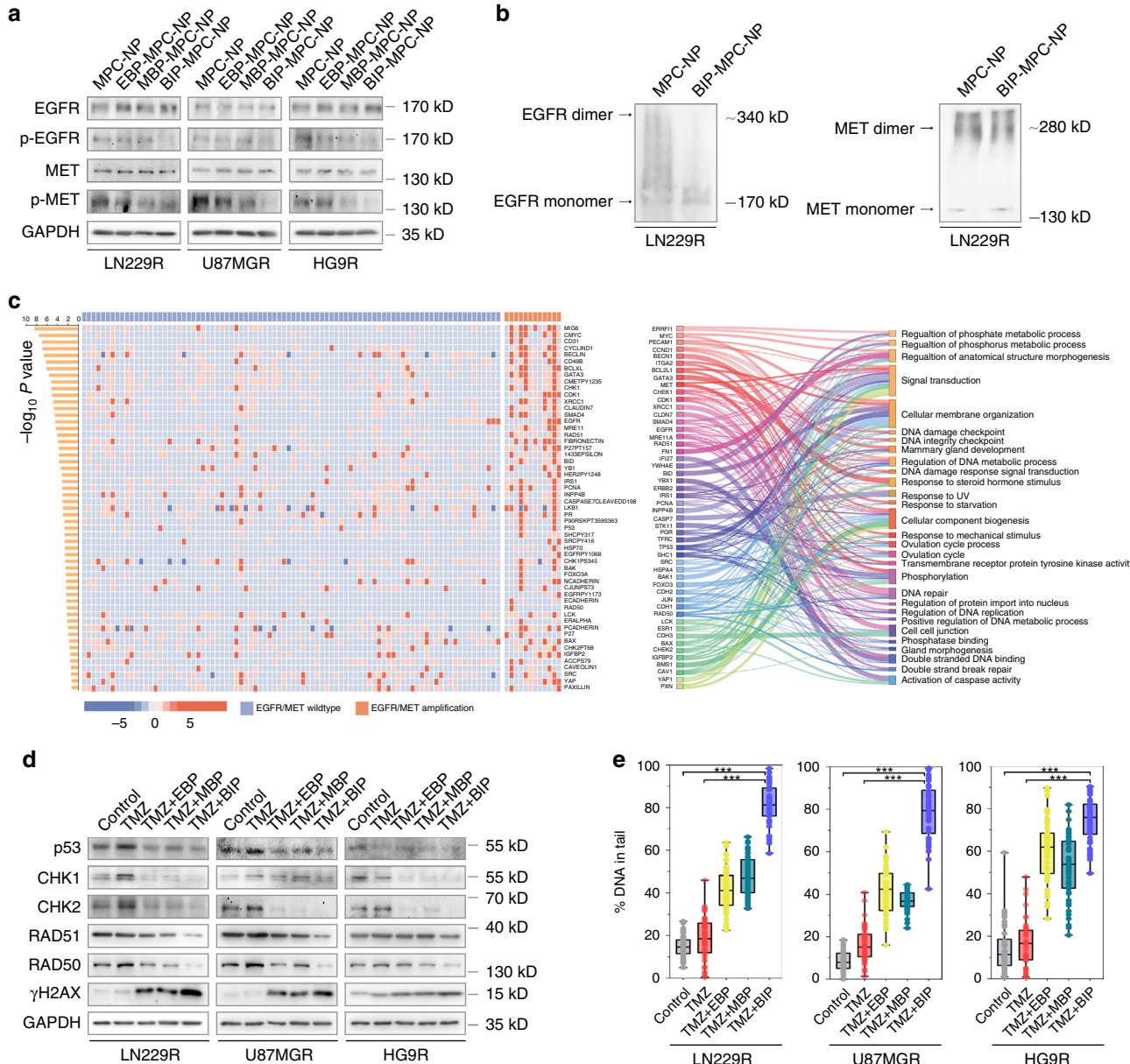

**Fig. 3 BIP-MPC-NP inhibits DNA damage repair via the mitigation of EGFR and MET signaling pathways. a** The EGFR, p-EGFR, MET and p-MET expression in TMZ-resistant glioma cells treated with different nanoinhibitors. **b** The levels of EGFR and MET dimers in TMZ-resistant glioma cells treated with BIP-MPC-NP. **c** The heatmap displayed the differential expressed proteins between GBM patients with low EGFR and MET copy numbers group and with high EGFR and MET copy numbers group. The Sankey diagram displayed the Gene Ontology of the differentially expressed proteins. **d** The inhibition of DNA damage repair modules expression and the increase of γH2AX expression with the treatments of TMZ and different nanoinhibitors in LN229R, U87MGR and HG9R. **e** The boxplots showing the statistics of comet assay of TMZ-resistant glioma cells with TMZ and different nanoinhibitors ($n = 50$). In the box plots, bounds of the box spans from 25 to 75% percentile, center line represents median, and whiskers visualize minimum and maximum of the data points. $P$ value is determined by Student's $t$-test. Significant results are presented as ***$P < 0.001$.

**BIP-MPC-NP targeting EGFR and MET enhances DNA damage.** Before elucidating the expression of EGFR and MET signaling pathways, the IC50 assays of U87MGR and HG9R were performed (Supplementary Fig. 7a). With nanoinhibitors as treatments on TMZ-resistant glioma cells, BIP-MPC-NP simultaneously attenuated p-EGFR and p-MET in cells with EGF and HGF incubation (Supplementary Fig. 7b) or without EGF and HGF incubation (Fig. 3a), and this attenuation was more significant than that of EBP-MPC-NP or MBP-MPC-NP group. BIP-MPC-NP was also found to significantly decrease the dimer formation of EGFR or MET, the key step of EGFR or MET activation[37,38], in the EGF- and HGF-dependent (Supplementary Fig. 7c) or -independent manners (Fig. 3b). The crosstalk signaling molecules of EGFR and MET pathways (p-AKT, p-p38, p-STAT3 and p-p65) were also mitigated by BIP-MPC-NP in TMZ-resistant glioma cells (Supplementary Fig. 7d).

Through the analysis of TCGA datasets, we observed that the upregulated proteins in the samples with high EGFR and MET copy numbers included EGFR, p-EGFR, p-MET and several DNA damage repair molecules such as CHK1, MERE11, RAD51, p53, RAD50 and p-CHK1 (Fig. 3c, Supplementary Table 3). These upregulated proteins were implicated in DNA damage repair, DNA damage response signal transduction and double-stranded break repair signaling pathways. We detected the expression

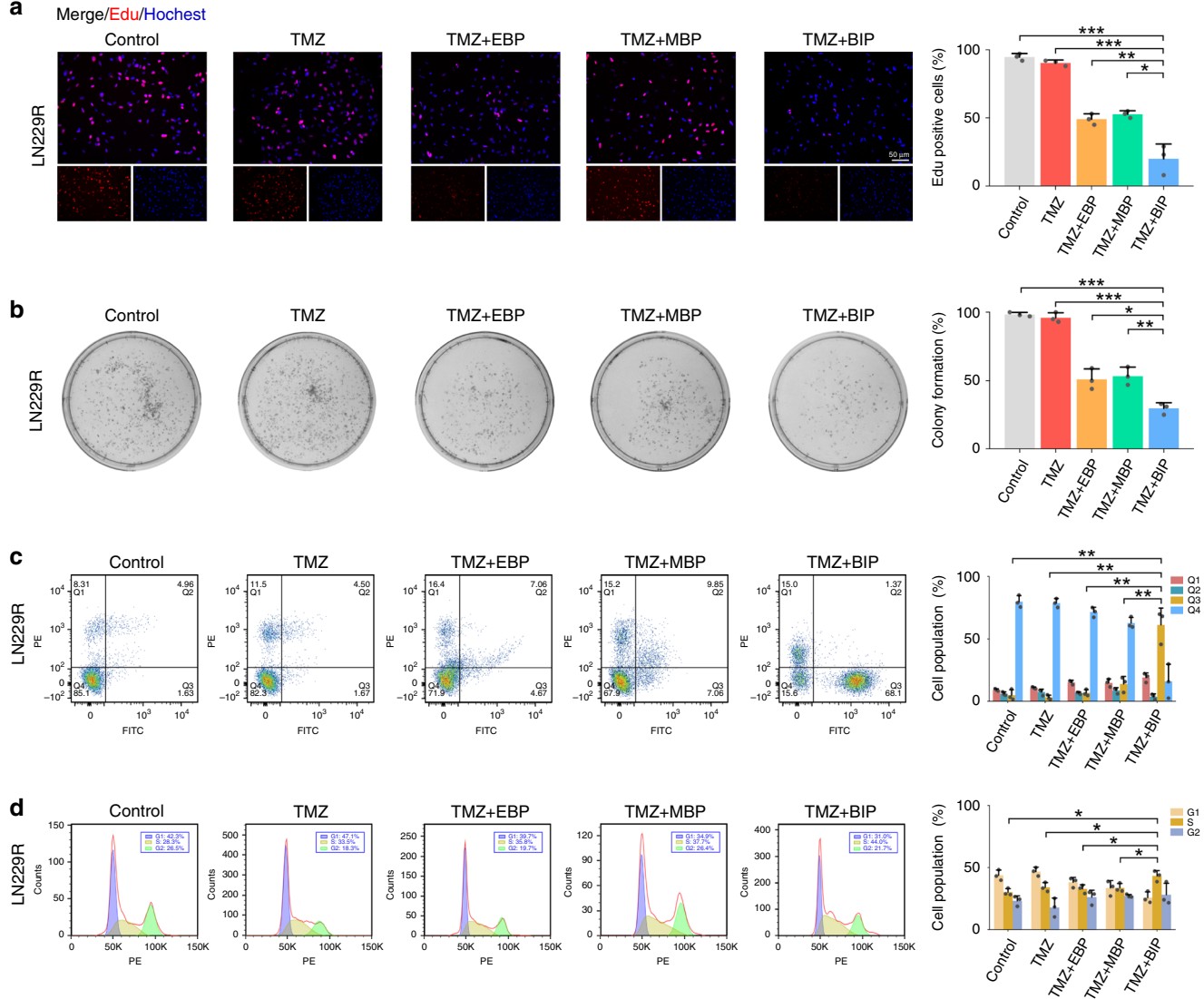

**Fig. 4 The TMZ sensitivity enhanced by BIP-MPC-NP in LN229R cells. a**, **b** The EdU assay and colony formation assay of LN229R with treatments of TMZ and different nanoinhibitors. The histogram displayed the statistics of EdU assay and colony formation assay, respectively ($n = 3$). Scale bar = 50 μm. **c**, **d** The cell apoptosis assay and the cell-cycle analysis were performed with flow cytometry in LN229R. The histogram displayed the statistics of apoptosis assay and the cell-cycle analysis, respectively ($n = 3$). The error bars represent the S.D. of three measurements. $P$ value was determined by Student's $t$-test. Significant results are presented as *$P < 0.05$, **$P < 0.01$, ***$P < 0.001$.

levels of the DNA damage repair molecules and observed that BIP-MPC-NP reduced the expression levels of CHK1, CHK2, RAD51, RAD50 and p53, and increased the expression levels of γH2AX induced by TMZ compared with EBP-MPC-NP or MBP-MPC-NP (Fig. 3d). Immunofluorescence (IF) assays further confirmed that CHK1, CHK2, RAD51, RAD50 and p53 expression were reduced in the TMZ + BIP (TMZ combined with BIP-MPC-NP) group compared with those in the TMZ + EBP (TMZ combined with EBP-MPC-NP) or TMZ + MBP (TMZ combined with MBP-MPC-NP) group in TMZ-resistant glioma cells (Supplementary Figs. 8–10). Comet assays also illustrated that BIP-MPC-NP enhanced the DNA damage induced by TMZ compared with EBP-MPC-NP or MBP-MPC-NP in TMZ-resistant glioma cells (Fig. 3e, Supplementary Fig. 11). To clarify the action of BIP-MPC-NP as an antitumor agent, we performed an 5-ethynyl-20-deoxyuridine (EdU) assay to investigate DNA replication status in TMZ-resistant glioma cells. The lower ability of DNA replication could be observed in the TMZ + BIP group than in the other groups (Fig. 4a, Supplementary

Fig. 12a, b). The clone formation assay and CCK-8 assay also validated that BIP-MPC-NP exhibited a greater proliferative inhibition effect induced by TMZ compared with EBP-MPC-NP or MBP-MPC-NP (Fig. 4b, Supplementary Fig. 13a, b). The rate of apoptotic cells and the population of cells arrested in S phase of the cell cycle in the TMZ+BIP group were increased compared with those in other groups (Fig. 4c, d, Supplementary Fig. 14a, b). These results demonstrated the potentiation of BIP-MPC-NP on TMZ-induced DNA damage and cytotoxicity effects in TMZ-resistant glioma cells.

**BIP-MPC-NP attenuates DNA damage repair by inhibiting TTP-mediated E2F1**. As downstream of the EGFR and MET cross-signaling pathway, E2F1 is responsible for the transcriptional regulation of many important modules governing a wide range of biological processes involved in the cellular response to DNA damage[39]. We thus employed the chromatin immunoprecipitation followed by next-generation sequencing (ChIP-seq)

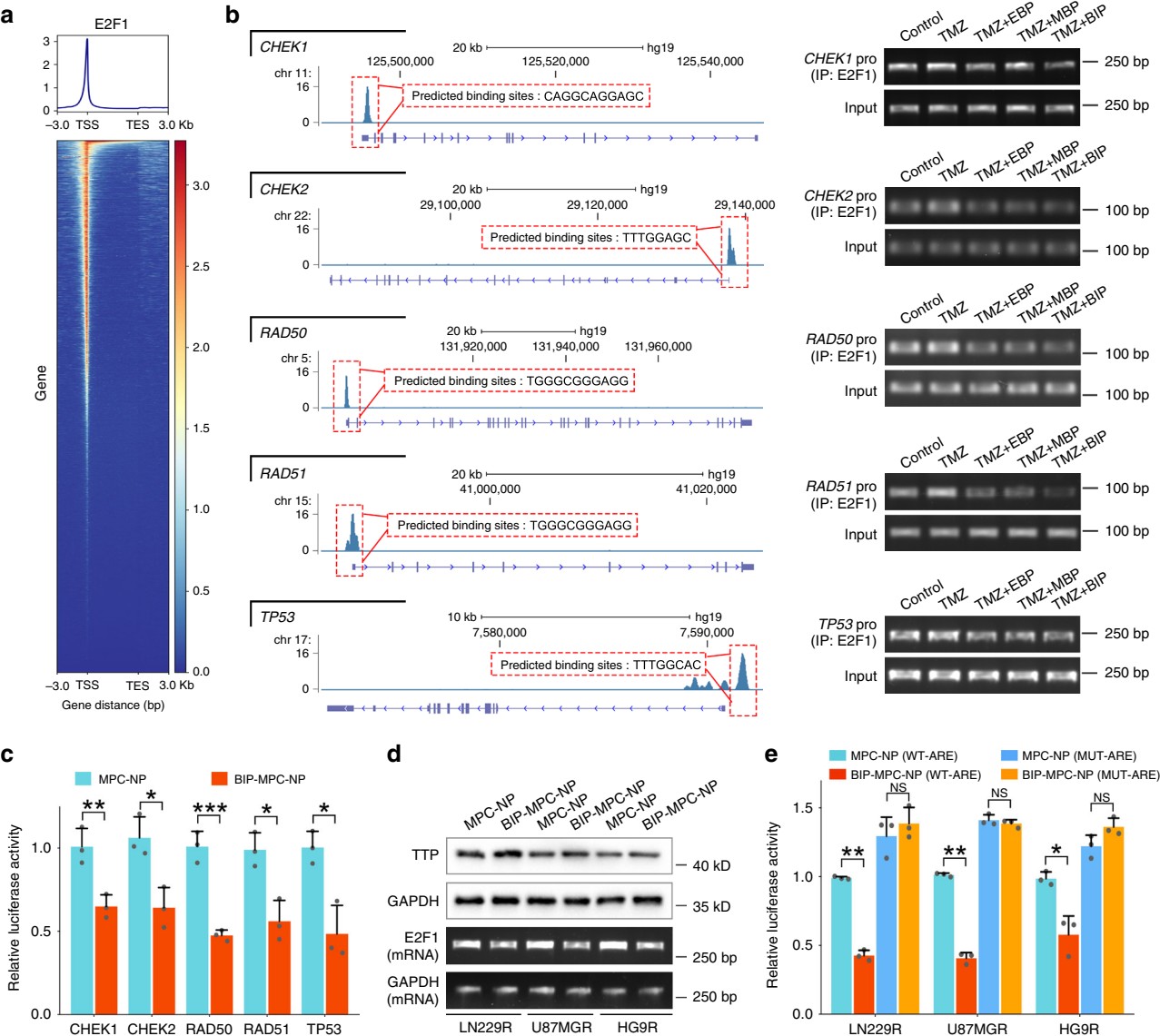

**Fig. 5 BIP-MPC-NP restrains E2F1-mediated DNA damage repair modules via the inhibitory effect of TTP. a** E2F1 binding sites within a region spanning ± 3 kb around TSS in the whole genome. **b** The signal peaks located in the promoter regions of *CHEK1*, *CHEK2*, *RAD50*, *RAD51* and *TP53* in E2F1 ChIP-seq data and the binding sites of E2F1 were predicted on JASPAR datasets. The agarose gel electrophoresis displayed the enrichments of E2F1 in the promoter regions of *CHEK1*, *CHEK2*, *RAD50*, *RAD51* and *TP53* of LN229R. **c** The luciferase reporter assay displayed the E2F1 transcriptional activity in the promoter regions of *CHEK1*, *CHEK2*, *RAD50*, *RAD51* and *TP53* in LN229R (*n* = 3). **d** The TTP protein expression and E2F1 mRNA expression in LN229R, U87MGR and HG9R treated with MPC-NP or BIP-MPC-NP. **e** The inhibitory effect of TTP on wild-type or mutant-type ARE motifs within the E2F1 mRNA 3′-UTR in LN229R cells with MPC-NP or BIP-MPC-NP treatments (*n* = 3). The error bars in **c** and **e** represent the S.D. of three measurements. *P* value is determined by Student's *t*-test. Significant results are presented as NS non-significant, *$P < 0.05$, **$P < 0.01$, ***$P < 0.001$.

datasets GSM2634759 and JASPAR datasets (http://jaspar. genereg.net) to predict the binding sites of E2F1 in the promoter regions of the *CHEK1*, *CHEK2*, *RAD50*, *RAD51* and *TP53* genes (Fig. 5a, b). Chromatin immunoprecipitation followed by polymerase chain reaction (ChIP-PCR) assays showed that BIP-MPC-NP could significantly downregulate the enrichment of E2F1 in the promoter regions of *CHEK1*, *CHEK2*, *RAD50*, *RAD51* and *TP53* genes compared with EBP-MPC-NP or MBP-MPC-NC in LN229R (Fig. 5b). We also observed that BIP-MPC-NP attenuated the E2F1 transcriptional activity in the promoter regions of these genes (Fig. 5c, Supplementary Fig. 15a). With the treatment of BIP-MPC-NP, the expression of E2F1 mRNA as well as protein was lower compared with that in the EBP-MPC-NP or MBP-MPC-NP group (Supplementary Fig. 15b, c), indicating that

the attenuation of EGFR and MET signaling pathways was responsible for E2F1 expression.

The analysis of the human E2F1 mRNA 3′-UTR revealed the presence of ARE motifs, which are usually recognized by RNA binding proteins such as TTP and target the mRNA for rapid degradation[40]. In our results, E2F1 mRNA was attenuated accompanied by increased TTP expression in cells treated with BIP-MPC-NP compared to those in the EBP-MPC-NP or MBP-MPC-NP groups (Fig. 5d). To illustrate that TTP protein could bind to ARE motifs in the 3′-UTR of E2F1 mRNA and promote mRNA degradation (Supplementary Fig. 16a), we cloned the wild-type (AUUUA) or mutant-type (AGUGA) ARE motifs derived from E2F1 mRNA 3′-UTR into the downstream of the luciferase reporter gene in the luciferase expression vector to

investigate the role of TTP in post-transcriptional regulation of E2F1 expression. Luciferase reporter activity was downregulated in cells containing wild-type ARE motifs following treatment with BIP-MPC-NP. In contrast, there was no significant difference in luciferase reporter activity in cells containing mutant ARE fragments (Fig. 5e). As downstream of the EGFR and MET signaling pathways, phosphorylated p38 (p-p38), which could be significantly attenuated by BIP-MPC-NP as previously described, is responsible for multiple biological processes, such as TTP downregulation. We employed SB203580, a p38-specific inhibitor, to further confirm the regulation of BIP-MPC-NP by decreasing E2F1 mRNA and protein levels via the upregulation of TTP by inhibiting the phosphorylation of p38. With the incubation of SB203580, p-p38 was attenuated with increasing TTP compared to that in cells treated with DMSO as a control group. In cells treated with SB203580, E2F1 mRNA and protein expression was synergistically decreased by the mitigation of p-p38 levels (Supplementary Fig. 16b), consistent with these results of BIP-MPC-NP treatment. We also validated that SB230580, similar to BIP-MPC-NP, led to decreased luciferase reporter activity in cells containing wild-type ARE fragments instead of in cells containing mutant-type ARE fragments. No significant difference in luciferase reporter activity was found in cells containing mutant ARE motifs (Supplementary Fig. 16c). These results demonstrated that BIP-MPC-NP could attenuate phosphorylation of p38 and reduce E2F1 expression mediated by TTP on ARE-containing mRNA to attenuate DNA damage repair in TMZ-resistant glioma cells.

**BIP-MPC-NP restrains TMZ-resistant glioma in vivo**. To expand our investigation to explore whether BIP-MPC-NP exhibited the effective improvement of TMZ sensitivity for TMZ-resistant glioma in vivo, we transplanted LN229R cells into 4-week-old female BALB/c nude mice. TMZ was intraperitoneally injected as 60 mg kg$^{-1}$ every day, and the therapeutic nanoinhibitors were intravenously administered once every other day after the third postoperative day. Magnetic resonance images (MRI) were collected on the 7th, 15th and 21st postoperative days, and the results showed that TMZ-resistant gliomas derived from LN229R cells were restrained by TMZ combined with BIP-MPC-NP (Fig. 6a). The hematoxylin–eosin (H&E) staining images were used to demonstrate the tumor sizes (Fig. 6a). At the 7th, 15th and 21st postoperative days, the volumes of the tumor xenografts increased exponentially and were measured as 2.1, 11.2 and 108.1 mm$^3$ in the control group, and those were determined as 2.0, 10.9 and 105.1 mm$^3$ in the TMZ+MPC group, respectively (Fig. 6b). In contrast, TMZ combined with BIP-MPC-NP revealed the most prominent inhibiting effect of tumor development, in which the volumes of tumors were determined to be 2, 3 and 12.6 mm$^3$ on the 7th, 15th and 21st days. Compared to the mouse models with a median survival of 23 days in the control group, those in the TMZ+MPC group achieved an almost unchanged median survival of 22 days (Fig. 6c). More significant therapeutic responses were recorded after injecting TMZ combined with BIP-MPC-NP, in which the median survival was extended to 47 days compared with that in the TMZ + EBP or TMZ + MBP groups (TMZ + EBP vs TMZ + BIP, $P < 0.01$, log-rank test; TMZ + MBP vs TMZ + BIP, $P < 0.01$, log-rank test). The TMZ + BIP group showed significantly reduced expression levels of p-EGFR and p-MET in glioma tissues by IF and IHC assays (Supplementary Fig. 17). The higher expression of TTP and γHAX, and the lower expression of E2F1, CHK1, CHK2, p53, RAD50 and RAD51 in glioma tissues were also observed in the TMZ + BIP group compared with other groups (Supplementary Figs. 18, 19). The safety of such a nanoinhibitor therapy had also been confirmed,

and neither the pathological changes in the visceral organs (Supplementary Fig. 20), hematologic toxicity (Supplementary Table 4) nor the significant impact on hepatic and renal functions (Supplementary Table 5) could be found. These results demonstrated that BIP-MPC-NP potentiated the TMZ-induced restriction on TMZ-resistant glioma xenografts by attenuating DNA damage repair via mitigation of the EGFR and MET signaling pathways (Fig. 7).

## Discussion

TMZ is demonstrated as the first-line chemotherapy drug in patients with glioma[41]. However, the survival rates still remain low from inherent TMZ resistance or relapse with chemotherapy-resistant gliomas[42,43]. Currently, there are few alternate treatment options for patients with TMZ-resistant gliomas, and adjuvant TMZ chemotherapy options are a popular area of intense research[33]. Clinical practices and experimental evidence have clearly revealed that biologic therapies are effective only in a certain percentage of GBM expressing the appropriate targets[4].

In the present study, whole-genome CNVs of glioma were detected in the TCGA datasets, and the most common amplification events on chromosome 7 (EGFR/MET) were found at high frequencies in GBM, consistent with previous reports[44,45]. Although histological classification is well established and is the basis of the WHO classification of CNS tumors, diffuse gliomas suffer from high intra- and inter-observer variability among grade II–IV tumors[46]. Recent molecular features studies have benefited from the availability of the datasets generated by TCGA and have related genetic, gene expression, and DNA methylation signatures with prognosis[44,47–50]. As shown in Brennan et al.'s work[44], at least one RTK was found to be altered in 67.3% of GBM, including EGFR (57.4%). MET alterations, including MET exon 14 skipping (METex14) and PTPRZ1-MET (ZM) fusion, in cell lines and xenografts demonstrated hyper-activation of the MET signaling pathway and acceleration of glioma proliferation[51].

Our results also showed that simultaneous knockdown of EGFR and MET could significantly downregulate the phosphorylation levels of the downstream proteins. It was a major characteristic in GBM that RTKs, including IGFR, HGFR (MET), FGFR, VEGFR and the EGFR family, were abnormally activated[6]. Upon activation by ligands, RTK signals would be transduced through two major downstream pathways, Ras/MAPK/ERK and Ras/PI3K/AKT[8], involved in the regulation of cancer chemoresistance[10,11]. EGFR is a member of the Erb family of RTKs regulating EGFR/NEAT1/EZH2 axis, and is critical for glioma cell growth and invasion[52]. EGFR activated the JNK-ERK1/2-AP-1 axis to induce Cx43 implicated in gap junctional intercellular communication among the resistant GBM cells[11]. MET, as an RTK, was an independent prognostic factor for TMZ chemotherapy[17]. Low expression levels of MET are prognostic for and predict the benefits of TMZ chemotherapy in GBM[17]. MET mediated the plasticity of endothelial cells to undergo mesenchymal transformation, rendering GBM resistant to TMZ[53]. Poor efficacy in the clinical treatment of cancers overexpressing EGFR may be due to the cross-talk between EGFR and MET[14,15]. The intricate network of cross-signaling involving EGFR and MET has also been reported in the past few years[54]. Targeting HGF/MET induces tumor cell apoptosis through cell-cycle arrest and DNA damage and suppresses tumor progression[18]. Thus, simultaneous targeting of both receptors could be an effective therapeutic strategy.

The use of nanoparticles in medicine has recently become an intensely studied field. Although nanoparticles carrying drugs have already reached the clinic, an important function to add to make nanoparticles smarter is active homing to the target

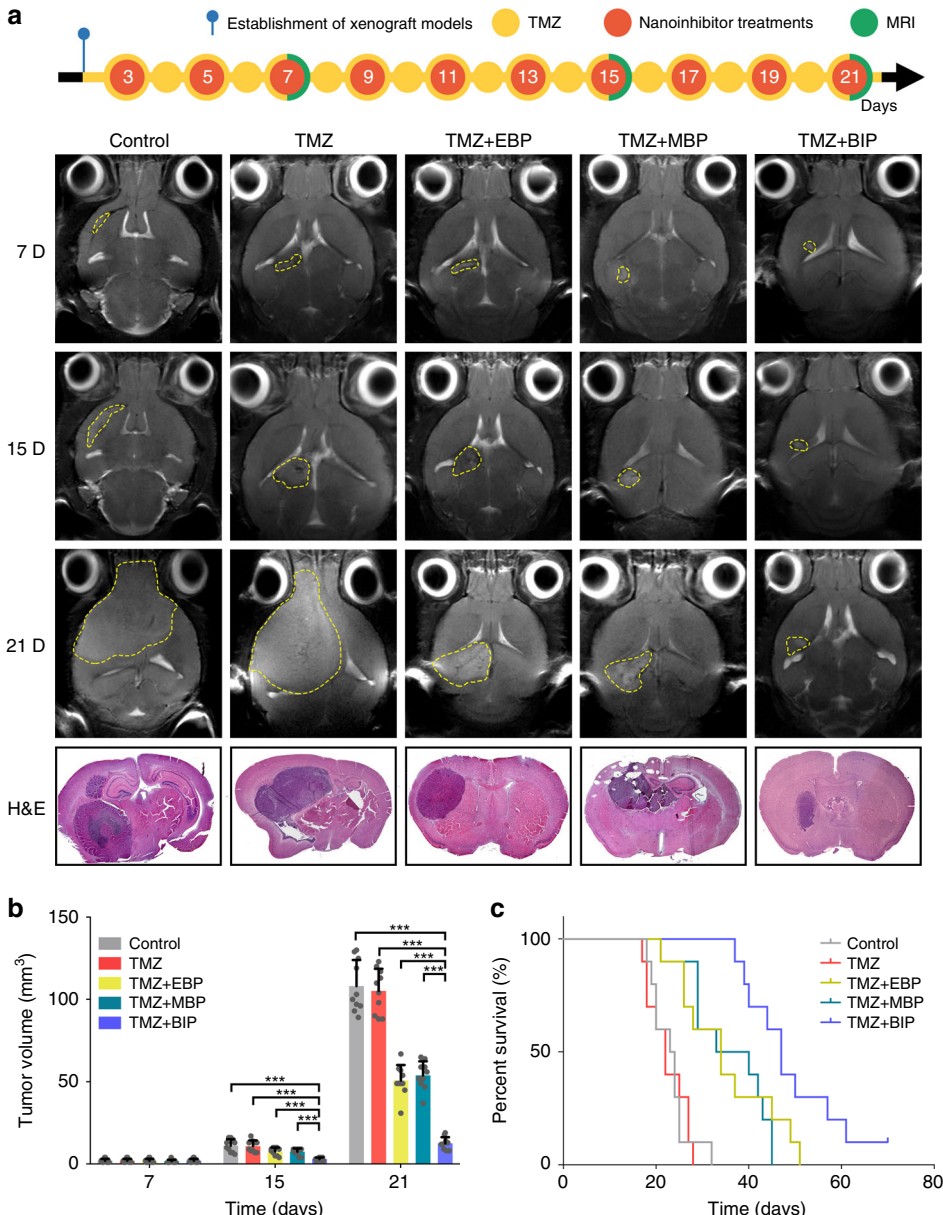

**Fig. 6 The therapeutic response of BIP-MPC-NP combined with TMZ in xenografts tumor derived from TMZ-resistant glioma cells. a** The diagram showed the process of the establishment of models, drug administration and MRI scan. T2-weighted MR images of mouse head at selected times post treatment with TMZ and different nanoinhibitors. The tumor sizes were visualized by H&E staining image. **b** Tumor volumes were measured on selected days post tumor implantation ($n = 10$). **c** Kaplan–Meier survival curves of mouse models after corresponding treatments ($n = 10$). $P = 4.33E-10$ for the four groups. Control vs TMZ, $P = 9.04E-01$; Control vs TMZ + EBP, $P = 4.22E-04$; Control vs TMZ + MBP, $P = 3.40E-04$; Control vs TMZ + BIP, $P = 4.00E-06$; TMZ vs TMZ + EBP, $P = 4.13E-02$; TMZ vs TMZ + MBP, $P = 6.20E-05$; TMZ vs TMZ + BIP, $P = 4.00E-06$; TMZ + EBP vs TMZ + MBP, $P = 1.57E-01$; TMZ + EBP vs TMZ + BIP, $P = 1.32E-02$; TMZ + MBP vs TMZ + BIP, $P = 4.82E-03$. The error bars in **b** represent the S.D. of ten measurements. $P$ value is determined by Student's $t$-test or log-rank test. Significant results are presented as $*P < 0.05$, $**P < 0.01$ or $***P < 0.001$.

tissue[55]. Peptides have the advantage of more easily diffusing into tissue and having low immunogenicity, yet they are prone to degradation and rapid clearance. Thus, peptides are often used in a stabilized form or as a modular targeting domain in a larger context[23]. Several approaches have been used to identify targeted peptides[23]. Inherbin3, an antagonist of epidermal growth factor (EGF)-EGFR signaling, inhibits EGF-induced EGFR phosphorylation, cell growth and migration in human tumor cell lines and suppresses tumor growth in a tumor xenograft model[25]. cMBP peptide targets ligand due to its high binding affinity to MET and application in tumor imaging[36]. Nanoinhibitors containing cMBP peptides conjugated on the G4 PEGylated dendrimer have been

shown to efficiently reduce the proliferation and invasion of human glioblastoma cells by blocking MET signaling with remarkably attenuated levels of phosphorylated MET and its downstream signaling proteins, such as pAKT and pERK1/2[28]. However, with the presence of BBB, glioma patients hardly benefit from immunological therapy[30]. Generally, the BBB formed by tightly packed cells lining the walls of vessels prevents harmful toxins and bacteria in the blood stream from entering the vital organ. What evolved as a life-saving defense, however, also blocks many molecules from reaching the brain, creating a major problem in treating brain tumors[29]. Based on the molecular structure of phosphatidycholine of cell membranes, a synthetic

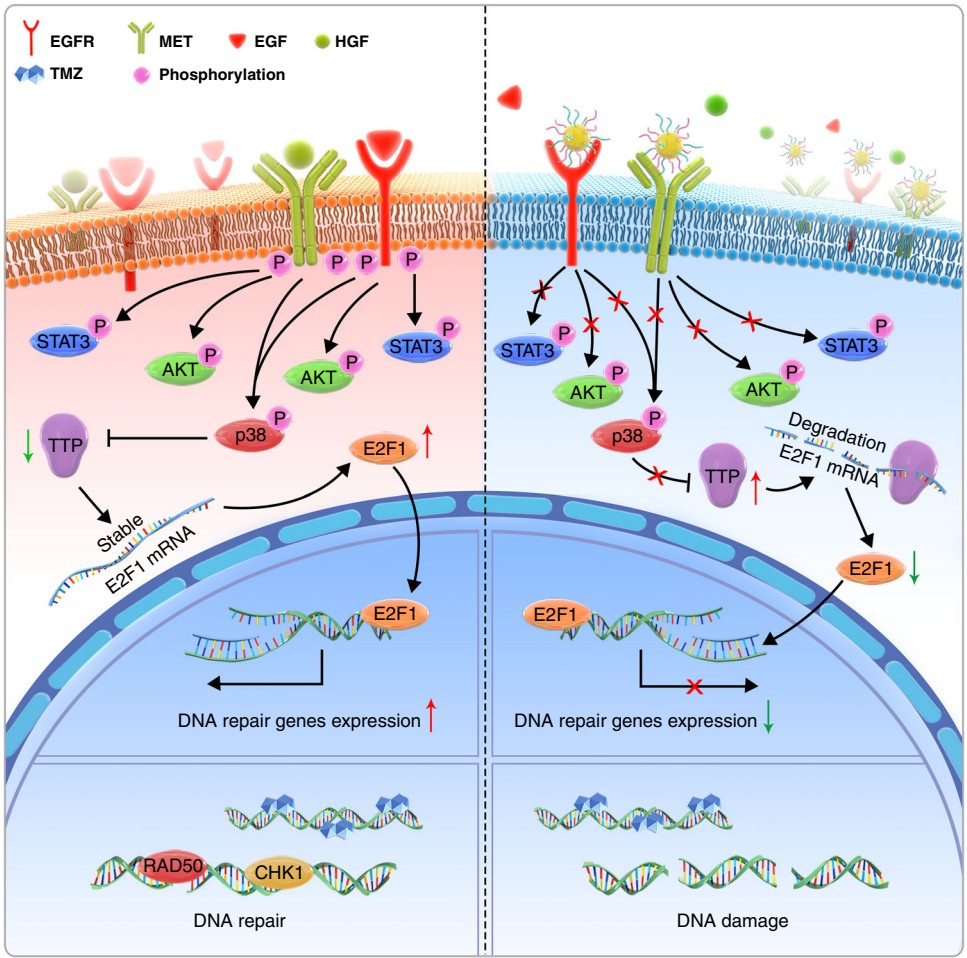

**Fig. 7 The mechanistic scheme of a dual functionalized BIP-MPC-NP in temozolomide-resistant glioma.** The mechanistic scheme of BIP-MPC-NP attenuating DNA damage repair to enhance TMZ sensitivity by simultaneously mitigating EGFR and MET activation in TMZ-resistant gliomas.

molecule, MPC, is synthesized to deliver drugs, antibodies or peptides to the central nervous system across the BBB[34,56]. In our study, we developed a nanoinhibitor, BIP-MPC-NP, that can simultaneously mitigate EGFR and MET activation by conjugating Inherbin3 and cMBP on the surface of NHS-PEG$_8$-Mal modified MPC nanoparticles. The fluorescence assays that presented the colocalization of factin and FITC-labeled nanoinhibitors confirmed the successful enrichment of nanoinhibitors on the glioma cell membrane. The transportation assays showed that BIP-MPC-NP had higher permeability on crossing BBB in comparison to BIP-NP. The effectiveness of delivering nanoparticles to brain tumors by the nanocapsules can be attributed to the MPC moiety. Higher average radiance was detected in mouse brain tissues treated with BIP-MPC-NP than with BIP-NP, which indicated that nanoparticles with MPC had a long circulative half-life and remarkable BBB penetration ability.

BIP-MPC-NP was found to significantly decrease the dimer formation of EGFR or MET, which were reported to be the key step of EGFR or MET activation[37,38], in an EGF- and HGF-dependent or independent manner. The EGFR and MET activation and the downstream of EGFR and MET signaling pathways, such as p-AKT, p-p38, p-STAT3 and p-p65 in TMZ-resistant glioma cells, were also mitigated by BIP-MPC-NP, as expected. AKT activation plays an important role in the acquisition of resistance to anticancer therapy. As a critical serine/threonine kinase in the EGFR and MET signaling pathway, AKT controls a myriad of cellular functions, including the promotion

of anti-apoptosis, proliferation and regulation of the cell cycle. AKT has also been implicated as a direct participant in the DNA damage response and repair induced by chemotherapy agents[57]. The induction of both p-STAT3 and MGMT was accompanied by acquisition of TMZ resistance in the GBM[58]. P65, a subunit of NF-kB, contributes to the pathogenicity of GBM by modulating many of the pathways, including the response to DNA damage and attenuating the efficacy of cytotoxic agents[59]. By activation of p38, the EGFR-mediated signaling pathways composed of DNA damage repair modules are clearly upregulated following the damage caused by drugs[60]. With BIP-MPC-NP treatments, the DNA damage repair modules were attenuated, and the cytotoxicity caused by TMZ was enhanced in TMZ-resistant glioma cells.

Mechanistically, E2F1 is an important transcription factor associated with EGFR and MET alterations[21,22]. Enforced E2F1 expression in advanced tumors and metastases of different types of cancers correlates with pronounced resistance toward therapy and poor patient prognosis[20,61]. Blocking EGFR signaling leads to disrupted E2F1 function[22], and direct knockdown of MET downregulated the expression of downstream molecules, including E2F1, in retinal pigment epithelia cells[21]. In the present study, the peak enrichment of E2F1 in the promoter regions of *TP53*, *CHEK1*, *CHEK2*, *RAD50* and *RAD51*, which were predicted on JASPAR datasets, was significantly downregulated in TMZ-resistant glioma cells with BIP-MPC-NP treatment. Luciferase reporter assays also showed that the E2F1 transcriptional activity on these genes was suppressed in TMZ-resistant glioma cells

treated with BIP-MPC-NP. When glioma cells were treated with BIP-MPC-NP, the E2F1 mRNA and protein expression levels were significantly downregulated.

The E2F1 mRNA 3′-UTR contains ARE motifs that can be recognized by RNA binding proteins[40]. ARE motifs are *cis*-acting mRNA decay determinants and participate in the regulation of post transcriptional processes[62]. Accumulating evidence suggests that TTP downregulates target mRNAs by contacting their ARE motifs[63]. In our results, BIP-MPC-NP downregulated E2F1 expression by the interaction between TTP and wild-type ARE motifs within the E2F1 mRNA 3′-UTR. TTP, one of the best characterized ARE binding proteins, could interact with mRNA and proteins, promote mRNA decay and operate as a translational regulator in maintaining homeostasis[63]. TTP could also be modulated by phosphorylated p38, which was found to play a major role in the regulation of mRNA stability[64]. As downstream of the EGFR and MET signaling pathways, phosphorylation of p38 is responsible for multiple biological processes[65]. In the current study, BIP-MPC-NP attenuated p-p38 expression and increased TTP expression. The p38-specific inhibitor SB203580 was also employed to further validate that inhibition of p-p38 decreased E2F1 mRNA and protein levels via the upregulation of TTP.

Intracranial xenograft models further elucidated that BIP-MPC-NP enhanced TMZ chemosensitivity via attenuating the EGFR and MET signaling pathways, including phosphorylation of p38 and reducing E2F1 expression accompanied by elevated TTP in TMZ-resistant gliomas. The expression of p-EGFR and p-MET decreased in the TMZ+BIP group compared with the other groups. BIP-MPC-NP had more effective inhibition of E2F1 expression and DNA damage repair modules (CHK1, CHK2, p53, RAD50 and RAD51) than other nanoinhibitors. The safety of such a therapeutic nanoinhibitor has also been confirmed. The major organs, including heart, liver, spleen, lung and kidney, were collected for histopathological analysis at the end of the nanoinhibitor treatment. Blood samples were collected for routine blood and serum enzyme assays. Neither pathological changes in the visceral organs nor significant impacts on hepatic and renal functions could be found.

Overall, we provided evidence that the activation and cross-talk of EGFR and MET signaling pathways contributed to TMZ resistance in glioma and developed a nanoinhibitor, BIP-MPC-NP, which could simultaneously mitigate EGFR and MET activation by conjugating EBP and MBP on the surface of NHS-PEG$_8$-Mal modified MPC nanoparticles. BIP-MPC-NP attenuated E2F1 expression by interactions between TTP and ARE motifs within the E2F1 3′-UTR and consequently downregulated DNA damage repair and enhanced TMZ chemosensitivity in TMZ-resistant glioma. These results demonstrated the promise of this nanoinhibitor as a feasible strategy in overcoming TMZ resistance in glioma.

## Methods

**Data collection**. The transcriptome expression profile, the CNV profile, the reverse phase protein arrays (RPPA) and clinical information data of patients diagnosed with glioma (lower grade glioma and glioblastoma) were obtained from TCGA (http://cancergenome.nih.gov/). The ChIP-seq data were obtained from the GSM2634759 data. The binding sites of E2F1 in the promoter regions of *CHEK1*, *CHEK2*, *RAD50* and *TP53* were predicted on JASPAR datasets (http://jaspar.genereg.net/). The gene expression profiling of parental and TMZ-resistant glioma cells was obtained from the GSE113510 dataset[33].

**Cell culture and transfection**. The patient-derived glioma cells were obtained from the glioma tissue of a female adult patient. Briefly, the glioma tissue was washed in phosphate-buffered saline (PBS) and minced into 1 mm$^3$. After enzymatically dissociated by 0.05% trypsin, the cells were suspended in MEM-α medium (Corning, Armonk, NY, USA) with 10% FBS (BD Biosciences, San Jose, CA, USA) and were recognized as HG9. Human glioma cells LN229 and U87MG

cells were purchased from the Chinese Academy of Sciences Cell Bank. These cells were authenticated using STR assay (Genetic Testing Biotechnology, Jiangsu, China). The LN229 and LN229R cells were cultured in DMEM/F12 (Corning, Armonk, NY, USA) medium with 10% FBS. The U87MG, HG9, U87MGR and HG9R cells were cultured in MEM-α medium with 10% FBS. The bEnd.3 cells were cultured in DMEM (Corning, Armonk, NY, USA) medium with 10% FBS. All cells were incubated at 37 °C in a humidified atmosphere with 5% CO$_2$ and were negative for mycoplasma contamination.

The cells were transfected with siRNAs by using Lipofectamine 2000 (Invitrogen, USA). Briefly, $5 \times 10^5$ cells were seeded in 6-well plates overnight and transfected with siRNAs targeting EGFR or MET (GeneChem, Shanghai, China). The validation of siRNAs was detected by Western blot.

**Establishment of TMZ-resistant cells**. The establishment process of TMZ-resistant LN229, U87MG and HG9 cells were consistent with our previous report[33]. Briefly, these cells were seeded into 96-well plates at 6000 cells per well, and the half maximal inhibitory concentration (IC50) of TMZ was evaluated. Then, TMZ was added to the cell culture medium at an IC50 1/50 concentration in 6-well plates. When the cells grew stably, the TMZ dose was increased. Each dose would be kept for 15 days until the end of the fifth month. The induced TMZ-resistant glioma cells were termed as LN229R, U87MGR and HG9R.

**Synthesis of the bare MPC-nanoparticles**. The bare MPC-nanoparticle were synthesized in a simple two-step process. The BSA was first conjugated with N-acryloxysuccinimide (NAS) to attach acryloyl groups onto its surface. The amount of NAS (10% in DMSO, m/v) used was at 120:1 molar ratio (NAS to protein), and the conjugation was kept at 4 °C for 2 h. Then, the solution was dialyzed against PBS buffer (10 mM, pH = 7.4) to remove any unreacted NAS. Acryloylated protein solutions were store at −20 °C for the following experiments. The acryloylated protein concentration was tuned to 1 mg mL$^{-1}$ by diluting with PBS buffer (10 mM, pH = 7.4). Polymerization was initiated in situ by the addition of tetramethylethylenediamine (TEMED) and ammonium persulfate (APS) and kept at 4 °C for 2 h, using 2-methacryloyloxyethyl phosphorylcholine (MPC) and N-(3-aminopropyl) methacrylamide (APM) as the monomers, and N, N′-methylenebisacrylamide (BIS) as the cross-linker. After the polymerization, the reaction mixture was dialyzed against PBS buffer (10 mM, pH = 7.4) to remove unreacted monomers and by-products. Subsequently, the solution was passed through a size-exclusion column (Sepharose 6B) to remove unreacted proteins. The molar ratio of native BSA:MPC:APM:BIS:APS was 1:2700:300:300:500. The mass ratio of APS:TEMED was 1:2.

**Synthesis of the BIP-MPC-nanoparticles**. To develop the BIP-MPC-nanoparticles, NHS-PEG$_8$-Mal (100 equiv.) was added to bare MPC-nanoparticles in PBS buffer (10 mM, pH = 7.4). The mixture was stirred at 4 °C for 3 h and then dialyzed in PBS buffer with a molecular weight cut-off (MWCO) of 7000 Da. Then the mixture of equimolar EBP and MBP with tris (2-carboxyethyl) phosphine (TCEP) was added to the NHS-PEG$_8$-Mal-modified MPC-nanoparticles in PBS buffer (10 mM, pH = 7.4). The mixture was stirred at room temperature overnight and then dialyzed in PBS buffer with a MWCO of 7000 Da. The synthesis of the control nanoparticles: EBP-MPC-nanoparticles and MBP-MPC-nanoparticles were similar to that of BIP-MPC-nanoparticles mentioned above. We just replaced EBP with equimolar MBP to prepare the MBP-MPC-nanoparticles and replaced MBP with equimolar EBP to prepare the EBP-MPC-nanoparticles, respectively.

**Characterization of the BIP-MPC-nanoparticles**. TEM measurements were performed with a Talos F200C electron microscope at an acceleration voltage of 200 kV. To prepare the TEM sample, 10 μL sample solution was dropped onto a carbon-coated copper grid for 10–15 min and blotted with filter paper to remove excess liquid. Then the sample was negatively stained with 2% uranyl acetate (5–10 μL) for 1–2 min, blotted again and air-dried before analysis on TEM. Fluorescence imaging of BIP-MPC-nanoparticles was analyzed using a confocal microscope. Element-mapping of BIP-MPC-nanoparticles was analyzed by transmission electron microscopy (JEM-2800). The size distribution and zeta potential of different nanoinhibitors were evaluated via a ZETAPALS/BI-200SM instrument in PBS buffer (10 mM, pH = 7.4) at room temperature. BIP-MPC-nanoparticles (100 μg mL$^{-1}$) was incubated at 37 °C in PBS buffer (10 mM, pH = 7.4) for 48 h. DLS measurements were performed for evaluating the stability of nanoinhibitors.

**The amount of peptide fragments in per nanoinhibitor**. The method of quantization to confirm the amount of different peptide fragments, where NR$^{BSA}$ is the number of residues of BSA, NR$^{PEP}$ is the number of residues of different peptide (e.g., MBP with 13 aa residues and EBP with 18 aa residues, the mixture of equimolar EBP and MBP with 31/2 aa residues), X is the rough number of peptide fragments. The concentration of nanoinhibiters including BSA and peptide content was confirmed via bicinchoninic acid (BCA) colorimetric protein assay (denoted as $C^{BSA+PEP}$) and BSA as the standard. We confirmed the concentration of BSA within nanoinhibiters by measuring the fluorescein$_{A495}$/protein$_{A280}$ (denoted as

$C^{BSA}$, the BSA molecules were prelabeled with FITC before encapsulation) following the calculation in Eq. (1):

$$\frac{NR^{BSA}}{NR^{BSA} + NR^{PEP} \times X} = \frac{C^{BSA}}{C^{BSA+PEP}}.$$

**Nanoinhibitors cell-adherance assay**. LN229R cells were collected and $1 \times 10^5$ cells were plated in 12-well plates overnight. The medium was replaced with fresh DMEM (supplemented with 10% FBS), and 30 min later, the cells were incubated with different nanoinhibitors (labeled with FITC). After 2-h further incubation, the culture medium was removed, and cells were washed three times with PBS buffer. The cellular adherence of nanoinhibitors was observed with an inverted fluorescence microscope (TCS SP8, Leica, Germany). The concentration of nanoinhibitors was 50 μg mL$^{-1}$. For quantitative analysis of cellular adherence, cells were seeded into 12-well plates at a density of $2 \times 10^5$ cells per well in 1 mL of fresh DMEM (supplemented with 10% FBS). After incubating for overnight, the culture medium of each well was replaced with 1 mL of fresh medium containing different nanoinhibitors. After 1-h further incubation, the culture medium was removed and cells were washed three times with PBS buffer and detached by 0.02% (w/v) EDTA and 0.25% (w/v) trypsin solution, and then dispersed in 0.5 mL of PBS for flow cytometric measurement. Cells treated with PBS were used as control. Data were analyzed under FlowJo software.

**Transportation assay and nanoinhibitors cell-adherance assay**. The bEnd.3 cells were seeded in the upper chambers (Corning, pore size: 0.4 μm) at a density of $2 \times 10^5$ cells per insert. After 2 weeks of incubation, the integrity of the barrier was evaluated via trans epithelial electric resistance (TEER) measurements. When the TEER value of bEnd.3 transwell reached ≥ 150 Ω cm$^2$, the transcytosis of nanoinhibitors across bEnd.3 monolayer was performed[66]. LN229R cells were collected and $1 \times 10^5$ cells were plated in the lower chambers overnight. Then, the in vitro BBB models were exposed to different FITC-labeled nanoinhibitors, including EBP-NP, MBP-NP, BIP-NP, MPC-NP, EBP-MPC-NP, MBP-MPC-NP and BIP-MPC-NP.

For cell-adherance assay, the medium in the lower chamber was collected for fluorescence detection after incubation. Then culture medium was removed, and cells were washed three times with PBS buffer. The cellular adherence of nanoinhibitors was observed with an inverted fluorescence microscope (TCS SP8, Leica, Germany). The concentration of nanoinhibitors was 50 μg mL$^{-1}$.

For quantitative analysis of cellular adherence, the culture medium of each well was replaced with 1 mL of fresh medium containing different nanoinhibitors after incubating for overnight. After 1-h further incubation, the culture medium was removed and cells were washed three times with PBS buffer and detached by 0.02% (w/v) EDTA and 0.25% (w/v) trypsin solution, and then dispersed in 0.5 mL of PBS for flow cytometric measurement. Cells treated with PBS were used as control. Data were analyzed under FlowJo software.

For investigating whether the nanoinhibitors could retain their function after penetrating through BBB, the LN229R cells were collected for immunefluorescence assay after incubation of FITC-labeled nanoinhibitors. The targeted adherence to EGFR and MET were detected after 5 h incubation. The expression of p-EGFR and p-MET were detected after 24 h incubation.

**In vivo distribution and imaging**. The mice were randomly divided into two groups (three mice per group) and intravenously injected with 100 μL of Cy5.5-labeled BIP-MPC-NP and BIP-NP, respectively (The concentration was 4 μM. BSA was used as the standard.). The distribution of the BIP-MPC-NP and BIP-NP were imaged using IVIS Lumina imaging system (IVIS Lumina II, PerkinElmerm, USA) at 5 h postinjection. The results were analyzed using Living Image 3.1 software (Caliper Life Sciences).

**The density and ratio of peptides on the nanoinhibitors**. The density of peptides on per nanoparticle was confirmed via Eq. (2):

$$Density = \frac{Number_{[pep]}}{\pi d^2_{[nanoparticle]}}.$$

The number$_{[pep]}$ represents the total number of peptides measured by BCA method and $d_{[nanoparticle]}$ stands for the diameter of particles detected by DLS (Supplementary Table 2). The conjugation ratio was obtained using fluorescent pre-labeled peptides, including FITC-labeled MBP and RhB-labeled EBP. The specific-UV absorption on per BIP-MPC-NP was measured and the standard curve was visualized in line plots.

**CCK-8 assay, EdU assay and colony formation**. For CCK-8 assay, a total of $5 \times 10^3$ cells were seeded in 96-well plates overnight. The nanoinhibitors were administered starting from the next day. For the detection of IC50, the OD450 values of different concentration were measured after 24 h. For the cell proliferation assay, cells were treated with TMZ along with nanoinhibitors every day.

The cell viability was evaluated by the Cell Counting Kit 8 (Dojindo, Japan) and was measured at OD 450 nm with the BioTek Gen5 system (BioTek, USA).

For EdU assay, cells with different treatments were seeded into tissue-culture treated slides (Nest, Rahway, NJ, USA) overnight and then incubated with 10 μM EdU (Ribobio, Guangzhou, China) for 24 h. The cells were fixed and labeled with Apollo 567 (Ribobio) and Hoechst according to the manufacturer's protocol.

For colony formation, a total of 500 cells were planted in 6-well plated and were incubated with different treatments for 14 days. Then cells were fixed with 4% paraformaldehyde and were stained with Giemsa agents.

**Comet assay**. For the comet assay, $4 \times 10^4$ cells with different treatments were collected and resuspended in 100 μL of low-melting-point agarose. Then the mono-suspension was cast on a microscope slide and was gelled at 4 °C for 20 min. The slides were then immersed in a solution (Trevigen, Gaithersburg, MD, USA) that cause the cells to lyse at 4 °C for 1 to 2 h. After lysis of the cells, the slides were washed in distilled water to remove all salts and immersed in electrophoresis solution. An electric field was applied (25 V, 300 mA) for 30 min. The slides were then neutralized, stained with 1× SYBRGreen (Sangon, Shanghai, China) and analyzed using a microscope.

**Cell cycle and apoptosis detection**. The apoptosis detection was performed using the FITC Annexin V Apoptosis Detection Kit I (BD Biosciences). A total of $1 \times 10^6$ cells were collected for each group and were stain with FITC and PI following the manufacturer's protocol.

For cell-cycle analysis, cells were fixed in ice-cold 75% ethanol for 12 h and were collected after centrifuging at 200×g for 10 min. After washed with ice-cold PBS twice, cells were resuspended in 500 mL of propidium iodide (BD Biosciences) staining buffer for 30 min at room temperature. After staining, the cells were analyzed by flow cytometry.

**RNA extraction and qRT-PCR**. The total RNA was extracted using TRIzol reagent (TaKaRa, Otsu, Japan). The cDNAs were synthesized with a PrimeScript RT reagent kit (TaKaRa, Otsu, Japan) and were determined using the SYBR Prime-Script RT-PCR Kit (Roche, Roswell, GA, USA) according to manufacturer's protocols. The quantitative real-time reverse transcription-PCR (qRT-PCR) data were analyzed using the $2^{-\triangle\triangle Ct}$ method. The primers used in qRT-PCR were listed in Supplementary Table 6. Uncropped scans of these blots are reported in Supplementary Fig. 21.

**Western blot assay**. Briefly, cells were scraped and then collected in RIPA buffer (Solarbio) with 1% protease inhibitors. After centrifugation at 17,800×g for 30 min at 4 °C, the total protein was collected, and the concentration was measured with the spectrophotometer (NanoDrop). All samples were subjected to sodium dodecyl sulfate polyacrylamide gel (EpiZyme Scientific) electrophoresis, and the gels were transferred onto PVDF membranes (Millipore, USA). The membranes were blocked in a 5% milk-TBST solution and incubated separately with primary antibodies (Supplementary Table 7). Following incubation with HRP-labeled secondary antibodies, the protein bands were detected with the SuperEnhanced chemiluminescence detection reagents (Applygen Technologies Inc) in a Chemi-DocTM MP Imaging System (BioRad).

For dimerization analysis, chemical cross-linking experiments were designed to examine the ability of EGF and HGF to induce receptor dimerization[67]. Cells treated with or without BIP-MPC-NP were seeded in 100-mm dishes and were harvested in 0.4 mL ice-cold lysis buffer (20 mM sodium phosphate, pH = 7.4; 150 mM NaCl; 1% Triton X-100; 5 mM EDTA containing protease and phosphatase inhibitor cocktails). The protein concentration in cell lysates was determined by a Lowry-based Bio-Rad assay (Bio-Rad Laboratories, Hercules, CA, USA). Aliquots of 10 mg of protein were stimulated with 150 nM EGF[67] and 100 ng mL$^{-1}$ HGF[68] at room temperature. After 30 min incubation, the cross-linking was induced by addition of 40 mM glutaraldehyde and was stopped with 0.2 M glycine (pH = 9) 1 min later. Next, loading buffer with 5% β-mercaptoethanol was added to the samples and heated for 5 min at 100 °C. The samples were electrophoresed on 6% polyacrylamide gels (SDS-PAGE), transferred onto PVDF membranes at 30 V overnight at 4 °C and analyzed by Western blot assay. Uncropped scans of these blots are reported in Supplementary Fig. 21.

**Immunofluorescence**. Cells were plated in 24-well tissue culture plates with smears (WHB-24-CS, Shanghai, China). The cells were stained using standard procedures overnight incubation at 4 °C. Primary antibodies (Supplementary Table 7) were diluted in 1% BSA in PBS. Cells were washed three times with PBS and then incubated with TRITC Phalloidin for 1 h at room temperature. Goat anti-rabbit IgG or goat anti-mouse IgG was used as secondary antibody. The nucleus was stained with DAPI (Sigma, USA) and visualized with a fluorescence microscope (Nikon C2, Tokyo, Japan).

For IF assays on glioma tissues, the slices were deparaffinized and rehydrated and the antigen was unmasked in standard procedure. The following steps were the same as IF assay procedure.

**The tissue slice distribution**. After injection of cyanine 5.5 (Cy5.5)-labeled BIP-MPC-NP or Cy5.5-labeled BIP-NP for 5 h, the tumor-bearing brains were removed and further fixed, followed by dehydration in 15% sucrose and 30% sucrose at 4 °C. Frozen coronal sections of 20 μm in thickness were prepared and processed for fluorescence imaging.

**Hematoxylin–eosin staining and immunohistochemistry**. The glioma tissues were fixed in 4% paraformaldehyde for 24 to 48 h and were dehydration and paraffin embedded tissue in standard procedure. For H&E staining, after deparaffinization and rehydration, the slides were stained in hematoxylin solution for 8 min and in eosin–phloxine solution for 30 s to 1 min. Then slides were mounted with xylene.

For IHC assay, after the slices were deparaffinized and rehydrated, the antigen was unmasked. Then the slides were incubated with primary antibodies, biotinylated secondary antibody, diaminobenzidine (DAB, ZSGB-BIO, ZLI9018) staining agents and hematoxylin in order.

Quantitative evaluation was performed by examining each section using at least 10 different high-power fields with the most abundant stained cells[69,70]. Two independent neuropathologists reviewed and scored each stained slide in a double-blinded fashion. The proportion of stained cells counts per field was used for statistical analysis. Staining was scored using a 4-point scale from "−" to "+++", with "−" if there was no staining or very little staining, "+" if less than 10% of cells stained positively, "++" if 10–30% of cells stained positively, and "+++" if more than 30% of cells stained positively[71]. The representative imaged field were determined by the average method.

**Chromatin immunoprecipitation (ChIP)**. ChIP assays were performed using the Millipore ChIP Assay Kit (17-295) and the anti-E2F1 antibody (Abcam, ab4070) following the manufacturer's protocol. Briefly, $4 \times 10^7$ cells were fixed by 1% formaldehyde for 15 min. Then, crosslinking was stopped by adding 0.125 M glycine for 5 min. Cells were resuspended in 0.5 mL of cell lysis buffer containing $1 \times$ protease inhibitor cocktail for 15 min, vortexed every 5 min and centrifuged at $800 \times g$ for 5 min, and then the supernatant was aspirated. DNA were resuspended in 0.5 mL of Nuclei Lysis buffer and sonicated on ice until chromatin fragments were ~250–500 bp in size, as detected through agarose gel electrophoresis. DNA was purified and rehydrated, and then the sample was analyzed by PCR. The ChIP-PCR products were detected with 1.5% agarose gel electrophoresis. Uncropped scans of these blots are reported in Supplementary Fig. 21. Sequences of the primers used for ChIP-PCR in this study are listed in Supplementary Table 6.

**Luciferase reporter assay**. Genomic DNA fragments of the human *CHEK1*, *CHEK2*, *RAD50*, *RAD51* and *TP53* genes, spanning from −3000 to +1 relative to the transcription initiation site (from upstream by 3000 bases to downstream by 1 base relative to the transcription initiation site), were generated by PCR and cloned into pGL3-Basic vectors.

Three oligonucleotides containing ATTTA motifs of the E2F1 mRNA 3′-UTR were PCR-amplified. The PCR products were cloned into the XhoI/NotI sites of the psiCHECK2 Renilla/firefly dual-luciferase expression vector (Promega). Point-mutated constructs in which the AUUUA was substituted with AGCGA were PCR amplified. E2F1 ARE motif1 (ATTTAATTTA to AGCGAAGCGA); E2F1 ARE motif2 (ATTTATTTA to AGCGAGCGA); E2F1 ARE motif3 (ATTTATTTA to AGCGAGCGA) (point mutation site underlined).

Lysates of the transfected cells were mixed with luciferase assay reagent (Promega) and the chemiluminescent signal was measured in a Wallac Victor 1420 Multilabel Counter (EG&G Wallac) according to the manufacturer's protocol.

**In vivo tumor inhibition**. Four-week-old female athymic BALB/c nude mice were purchased from Beijing Vital River Laboratory Animal Technology Co., Ltd. (Beijing, China) and were randomly divided into five groups (ten mice per group). A total of $3 \times 10^5$ LN229R cells per mouse were stereotactically injected into the brain. After 3 days of the surgery, the mice were treated with TMZ (60 mg kg$^{-1}$ everyday) by intraperitoneal injection every day. EBP-MPC-NP (4 μM), MBP-MPC-NP (4 μM) or BIP-MPC-NP (4 μM) was injected intravenously every other day after the surgery. The intracranial tumors were measured with Magnetic Resonance Imaging 9.4T horizontal Bruker magnets (Bruker Corporation, Beijing, China) and the consecutive sections (0.7 mm) were obtained at the 7th, 15th and 21st day after tumor implantation. The craniocaudal diameter (*dcc*), the anteroposterior diameter (*dap*) and the largest lateral diameter (*dl*) were measured by MRI. Diameter-based measurements were computed to calculate the volume (*V*) via Eq. (3):

$$V = \frac{dcc \times dap \times dl \times \pi}{6}.$$

The brain tissues and other major organs were embedded in paraffin and sectioned at a thickness of 2 μm for IHC assay and H&E staining. The blood samples were collected at the same time for serum enzymes assays at the end of different nanoinhibitor treatments. All procedures were approved by the Committee on the Ethics of Animal Experiments of Harbin Medical University.

**Statistical analysis**. Student's *t*-test was used to assess differences in the variable groups. Univariate survival analyses were conducted using the Kaplan–Meier

curves and the log-rank test using the GraphPad Prism 7 software. Comparisons between quantitative evaluation of IHC assays were performed via the Chi-squared test. Gene Ontology (GO) analysis was carried out with differential expressed genes (fold change >1.2) on DAVID website (http://david.ncifcrf.gov/). R version 3.3.2 with the extension package pheatmap were used to produce figures. Circos software was used for the circos plot. Integrative Genomics Viewer (IGV) was used for the visualization of ChIP-seq data. Statistical values of $P < 0.05$ were significant.

**Study approval**. We have complied with all relevant ethical regulations for animal testing and research. All experiments were approved by the Institutional Review Board at the Second Affiliated Hospital of Harbin Medical University and were in accordance with the principles expressed at the Declaration at Helsinki. All animal experiments were performed according to Health guidelines of Harbin Medical University Institutional Animal Use and Care Committee.

The informed consents were obtained from patients involved in this study, and the study protocol was approved by the Clinical Research Ethics Committee of the Second Affiliated Hospital of Harbin Medical University.

**Reporting summary**. Further information on research design is available in the Nature Research Reporting Summary linked to this article.

## Data availability
The transcriptome expression profile data, the CNV data, the RPPA data and clinical information data of patients diagnosed with glioma (lower grade glioma and glioblastoma) referenced during the study are available in a public repository from the TCGA data portal (http://cancergenome.nih.gov/). The ChIP-seq data were obtained from the GSM2634759 data and the gene expression profiling of parental and TMZ-resistant glioma cells was obtained from the GSE113510 dataset (https://www.ncbi.nlm.nih.gov/geo). All the other data supporting the findings of this study are available within the article and its supplementary information files and from the corresponding author upon reasonable request. A reporting summary for this article was available as a Supplementary Information file.

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

## Acknowledgements

We thank all the participants, investigators, and staff for expert intellectual and technical advice for our studies and useful discussion for the manuscript. This study was supported by: 1. National Natural Science Foundation of China (Nos. 81702972, 81874204, 81572701, 81772666, 81972817); 2. China Postdoctoral Science Foundation (2018M640305, 2019M660074, 2019M660975); 3. Chinese Society of Neuro-oncology, CACA Foundation (CSNO-2016-MSD12); 4. Special Fund Project of Translational Medicine in the Chinese-Russian Medical Research Center (No. CR201812); 5. Heilongjiang Postdoctoral Science Foundation (LBH-Z18103); 6. Heilongjiang Health and Family Planning Commission Foundation (2017-201); 7. National Key Research and Development Programs of China (2018YFA0209700).

## Author contributions

C.J., Y.L. and J.C. designed the experiments. X.M. and Y.Z. performed the experiments. X.M. and Y.Z. analyzed the data. J.C., Y.Z. and X.M. wrote the manuscript. B.H., C.Z., Y.Z., Z.L., P.W. and T.Q. contributed materials and analysis tools. All authors revised the manuscript.

## Competing interests

The authors declare no competing interests.
