## [Peer Review File · Nature Communications]

Reviewers' comments:

Reviewer #1 (Expertise: Nanoparticles, peptide delivery to brain, Remarks to the Author):

In this manuscript, the authors developed a dual functionalized brain-targeting nanoinhibitor, BIP-MPC-NP, via conjugating Inherbin3 and cMBP on the surface of NHS-PEG8-Mal modified MPC-nanoparticles to simultaneously mitigate EGFR and MET activation. They found the overactivation of EGFR and MET signaling pathways in TMZ resistant gliomas, and researched the mechanism that BIP-MPC-NP reduced E2F1 expression by the interaction between TTP and ARE motifs via the inhibition of the p38 phosphorylation, decreased DNA damage repair modules and enhanced the TMZ sensitivity. Although positive results were observed, it is still premature at current stage. Several comments need to address before further consideration.

1. The BIP-MPC-NP should penetrate through BBB and then recognize EGFR and MET for sensitizing glioma cells to temozolomide. However, the penetration is involved internalization into endosomes and escape from the endosomes, while there are fruitful enzymes in the endosomes. Therefore, authors should evaluate the binding affinity of the BIP-MPC-NP with targets after penetrating through BBB.
2. Authors should report the density and ratio of EBP and MBP on the nanoparticles.
3. In Figure 3e, which kind of cells was used?
4. In localization study, authors only evaluated U87MGR cells (Fig 4a), but in IC50 assay, authors only evaluated LN229R cells (Fig S5a). Why did authors not use same cell line or all the three cell lines?
5. In figure 4c, the higher concentration of BIP-MPC-NP showed higher penetration efficiency. How to explain the result? Normally, higher concentration would result in lower penetration efficiency due to the saturation of receptors or transporters.
6. Authors evaluated the functions of BIP-MPC-NP by directly incubation with glioma cells. However, as BBB is an important barrier for glioma treatment, authors should evaluate the functions of BIP-MPC-NP in BBB and glioma cell co-culture model as widely used in other papers. It is not sure that the BIP-MPC-NP retained their function after penetrating through BBB.
7. In vivo imaging is not enough for evaluating the BBB penetration. Authors should provide tissue slice distribution to make sure the particles were internalized into brain tissue or cells.
8. Why was the temozolomide intraperitoneally injected? p.o. or i.v. is more suggestive to mimic the clinical treatment.
9. Which kind of cells was used for in vivo treatment study?
10. The in vivo distribution of the nanoparticles should be performed in glioma bearing mice.
11. Please add the detailed particle size and PDI of BIP-MPC-NP and MPC-NP separately.

12. Scale bars should be added to Figure 3d.

Reviewer #2 (Expertise: Glioma models, therapy, Remarks to the Author):

This manuscript by Meng et al, reports the development and implementation of dual-functionalized brain targeting nano-inhibitors in glioma models that exhibit resistance to TMZ. The authors elected to target EGFR and MET signaling pathways. The authors have provided substantial data, which are of high quality. The main concern with the overall aim of this work is that the authors have not compared the response of the TMZ resistant models vs the TMZ sensitive models. Their overall premise lies on this distinction, thus, it is critical that the authors show differential responses to the treatment utilizing these two model systems.

The authors should also provide histological data in relation to the size of the tumors at the time of treatment. This is critical as it would be important to demonstrate the presence of a tumor mass at the time the treatment starts.

The authors also need to provide neuropathological data from the brain of the animals that were treated and died of tumor burden, and compare the neuropathology with control, non-treated animals.

An analysis of immune cellular infiltration by immunohistochemistry should also be presented.

DNA damage should be demonstrated by the presence of γ H2AX foci in vivo, in response to the treatment and the foci should be quantified.

In addition, this paper is very difficult to read, the figure legends are poorly described and lack details required to follow content.

The paper would need to be thoroughly re-written to clarify the way the data is presented and also provide details related to the figures presented.

Reviewer comments and responses:

Response to Reviewer #1:

1. The BIP-MPC-NP should penetrate through BBB and then recognize EGFR and MET for sensitizing glioma cells to temozolomide. However, the penetration is involved internalization into endosomes and escape from the endosomes, while there are fruitful enzymes in the endosomes. Therefore, authors should evaluate the binding affinity of the BIP-MPC-NP with targets after penetrating through BBB.

Response: Thank you for your valuable comments. In the revised manuscript, we have evaluated the binding affinity of BIP-MPC-NP after penetrating through BBB by fluorescence and the function of BIP-MPC-NP by immunofluorescence.

In details, we have plated bEnd.3 (1×10^5 cells/well) in the upper chambers to establish a blood-brain barrier (BBB) model and temozolomide (TMZ) resistant glioma cells (LN229R) in the lower chambers for evaluating the binding affinity of BIP-MPC-NP. BIP-MPC-NP was administered in the supernatant of the upper chambers. The binding affinity of BIP-MPC-NP and TMZ resistant glioma cell was assessed by fluorescence (data was shown in the revised Figure 2a). We have also performed immunofluorescence to evaluate the biological function of BIP-MPC-NP in LN229R. After penetrating through BBB, BIP-MPC-NP could bind with TMZ resistant glioma cells targeting EGFR and MET respectively (data was shown in the revised Figure 2d), and attenuate the phosphorylation of both EGFR and MET (data was shown in the revised Figure 2e). With these experiments, we updated the *Method* section and *Results* section as following:

Methods (revised part):

Transportation assay and Nanoinhibitors cell-adherence assay

The bEnd.3 cells were seeded in the upper chambers (Corning, pore size: 0.4 μm) at a density of 2×10^5 cells/insert. After two-weeks of incubation, the integrity of the barrier was evaluated via trans epithelial electric resistance (TEER) measurements. When the TEER value of bEnd.3 transwell reached $\geq 150 \Omega \cdot \text{cm}^2$, the transcytosis of nanoinhibitors across bEnd.3 monolayer was performed¹. LN229R cells were collected and 1×10^5 cells were plated in the lower chambers overnight. Then, the *in vitro* BBB models were exposed to different FITC-labeled nanoinhibitors, including NP, EBP-NP, MBP-NP, BIP-NP, MPC-NP, EBP-MPC-NP, MBP-MPC-NP and BIP-MPC-NP.

For cell-adherence assay, the medium in the lower chamber was collected for fluorescence detection after incubation. Then culture medium was removed, and cells were washed three times with PBS buffer. The cellular adherence of nanoinhibitors was observed with an inverted fluorescence microscope (TCS SP8, Leica, Germany). The concentration of nanoinhibitors was 50 $\mu\text{g}/\text{mL}$.

For quantitative analysis of cellular adherence, the culture medium of each well was replaced with 1 mL of fresh medium containing different nanoinhibitors after incubating for overnight. After 1-hour further incubation, the culture medium was removed and cells were washed three times with PBS buffer and detached by 0.02% (w/v) EDTA and 0.25% (w/v) trypsin solution, and then dispersed in 0.5 mL of PBS for flow cytometric measurement. Cells treated with PBS were used as control. Data were analyzed under FlowJo software.

For investigating whether the nanoinhibitors could retain their function after penetrating through BBB, the LN229R cells were collected for immunofluorescence assay after incubation of FITC-labeled nanoinhibitors. The targeted adherence to EGFR and MET were detected after 5 hours incubation. The expression of p-EGFR and p-MET were detected after 24 hours incubation.

Results (revised part):

To elucidate the appropriate concentration of nanoinhibitors administered on cells, the IC₅₀ assays were investigated after 24-hour treatments in LN229R and we chose 4 μM as the concentration for further experiments (Supplementary Figure 6a). Then different nanoparticles were labeled with FITC (green) and in vitro fluorescence images showed the localization of EBP-MPC-NP, MBP-MPC-NP and BIP-MPC-NP on the surface of LN229R cells (red), while the MPC-NP without peptides showed negligible colocalization (Figure 2a). Flow cytometric analysis showed that compared with cells treated with PBS or MPC-NP, those treated with EBP-MPC-NP, MBP-MPC-NP or BIP-MPC-NP had stronger fluorescent intensities (Figure 2b), indicating that these nanoinhibitors could bind to the cell surface efficiently. With 4-hour incubation, 5.2% of BIP-MPC-NP (4 μM) penetrated through the bEnd.3 layer. (Figure 2c, Supplementary Figure 6b). After penetrating BBB model, the nanoinhibitors also had strong fluorescent intensities (Supplementary Figure 6c) and the binding affinity targeting EGFR and MET (Figure 2d). The expression of p-EGFR and p-MET were simultaneously attenuated by BIP-MPC-NP penetrating BBB model (Figure 2e).

Revised Figure 2:

2. Authors should report the density and ratio of EBP and MBP on the nanoparticles.

Response: Thank you for your valuable comments.

The density of peptides on per nanoparticle was confirmed via formula, $Density = (Number[pep]) / (\pi d[nanoparticle]^2)$, in which $number[pep]$ represents the total number of peptides measured by BCA method and $d[nanoparticle]$ stands for the diameter of particles detected by DLS. The conjugation ratio was obtained using fluorescent pre-labeled peptides, including FITC-labeled MBP and RhB-labeled EBP. The specific-UV absorption on per BIP-MPC-NP was measured and the standard curve was visualized in line plots. The *Methods* section was revised as following:

Methods (revised part):

The density and ratio of peptides on the nanoinhibitors

The density of peptides on per nanoparticle was confirmed via formula, $Density = (Number[pep]) / (\pi d[nanoparticle]^2)$, in which $number[pep]$ represents the total number of peptides measured by BCA method and $d[nanoparticle]$ stands for the diameter of particles detected by DLS (revised *Supplementary Table 5*). The conjugation ratio was obtained using fluorescent pre-labeled peptides, including FITC-labeled MBP and RhB-labeled EBP. The specific-UV absorption on per BIP-MPC-NP was measured and the standard curve was visualized in line plots.

3. In Figure 3e, which kind of cells was used?

Response: Thank you for your comments. In the Figure 3e, the fluorescence images demonstrated the co-localization of FITC-MBP and RhB-EBP for each BIP-MPC-nanoparticle. No cells were used in this fluorescence assay. In the revised manuscript, we moved the Figure 3e to revised *Figure 1e*. We have updated the description in the Results section and Figure legend.

Revised Figure 1e:

Results (revised part):

Fluorescence images presented the co-localization of FITC-MBP and RhB-EBP for each BIP-MPC-NP (Figure 1e).

Figure legend (revised part):

Figure 1. The physical characteristics of nanoinhibitors.

(e) Fluorescence images presenting the co-localization of FITC-MBP and RhB-EBP for each BIP-MPC-nanoparticle.

4. In localization study, authors only evaluated U87MGR cells (Figure 4a), but in IC50 assay, authors only evaluated LN229R cells (Figure S5a). Why did authors not use same cell line or all the three cell lines?

Response: Thank you for your valuable comments. In the original manuscript, we have evaluated the IC₅₀ of all three cell lines. However, we did not show them in the figures since the IC₅₀ of U87MGR and HG9R were higher than that of LN229R. In the revised manuscript, we provided all data of IC₅₀ in the revised *Supplementary Figure 6a* and revised *Supplementary Figure 7a*.

Revised Supplementary Figure 6a:

Revised Supplementary Figure 7a:

5. In figure 4c, the higher concentration of BIP-MPC-NP showed higher penetration efficiency. How to explain the result? Normally, higher concentration would result in lower penetration efficiency due to the saturation of receptors or transporters.

Response: Thank you for your valuable comments in presenting the result of penetration efficiency. Mediated by nicotinic acetylcholine receptor (nAChRs) and choline transporter (ChTs), the NPs coated with PMPC can be transported across the BBB and delivered into brain, which has been confirmed in our original work. Furthermore, when the culture medium was supplemented with the MPC monomer (100 mM), the penetration efficiency was reduced². As the reviewer mentioned, we observed the analogous results in higher concentration (e.g., 8 µM) in our experiments. With 5 hours incubation, 7.1% of BIP-MPC-NP (6 µM) penetrated through the bEnd.3 layer, which showed higher efficiency compared to 4 µM nanoparticles (5.2%). While the lower penetration efficiency was observed when the concentrations of BIP-MPC-NP were

exceeded 6 μM . However, based on the IC_{50} assay, 4 μM should be chosen as the appropriate concentration.

We didn't show the full data in the original manuscript, owing to the limitation of high concentration in practical application process. In the revised *Supplementary Figure 6b*, we provided the whole experimental results to back up the concerns of reviewer and improved the quality of our work.

In the revised manuscript, we have updated the *Result* section to compare the penetration efficiency of nanoinhibitors with MPC and that of nanoinhibitors without MPC at the concentration of 4 μM (revised *Figure 2c*).

Revised Supplementary Figure 6b:

Revised Figure 2c:

6. Authors evaluated the functions of BIP-MPC-NP by directly incubation with glioma cells. However, as BBB is an important barrier for glioma treatment, authors should evaluate the functions of BIP-MPC-NP in BBB and glioma cell co-culture model as widely used in other papers. It is not sure that the BIP-MPC-NP retained their function after penetrating through BBB.

Response: Thank you for your valuable comments. To evaluate whether the BIP-MPC-NP retained their function after penetrating through BBB, we estimated the binding affinity of BIP-MPC-NP targeting EGFR and MET, and evaluated p-EGFR and p-MET of glioma cells in BBB model *in vitro* by immunofluorescence. After penetrating BBB model, BIP-MPC-NP could still bind with EGFR and MET (revised *Figure 2d*). The expression of p-EGFR or p-MET was

down-regulated with BIP-MPC-NP treatment in BBB model (revised **Figure 2e**). These data demonstrated that BIP-MPC-NP retained their function after penetrating through BBB.

Revised Figure 2d-e:

7. In vivo imaging is not enough for evaluating the BBB penetration. Authors should provide tissue slice distribution to make sure the particles were internalized into brain tissue or cells.

Response: Thank you for your valuable comments. In the revised manuscript, we provided the tissue slice distribution to confirm that the nanoinhibitors could penetrate the BBB and are internalized into brain tissues. In the brain tissues and glioma tissues of nude mice, we would perform *ex vivo* fluorescence to show the distributions of BIP-MPC-NP and BIP-NP (revised **Supplementary Figure 6f**).

We have also added the description of the tissue slices and more details in the **Results** section and the **Methods** section.

Revised Supplementary Figure 6f:

Results (revised part):

To further evaluate the capability of the nanoinhibitors in crossing BBB *in vivo*, we employed tumor-bearing mice to perform this study. Cy5.5-labeled BIP-MPC-NP and BIP-NP were injected via the tail vein at a dosage of 100 μ L (4 μ M). The mice were then sacrificed, and the brains were harvested for *ex vivo* imaging 5 hours post the injection. According to the results (Figure 2f-h, Supplementary Figure 6d-e), the fluorescence signal of Cy5.5-labeled BIP-MPC-NP was observed clearly in the mouse brains and showed a 3.2-fold higher increase than BIP-NP, indicating effective BBB permeability. The *ex vivo* fluorescence images of the sliced tumor-bearing brains tissues showed the enhanced distribution of Cy5.5-labeled BIP-NP with MPC (Supplementary Figure 6f). These results demonstrated that the nanoinhibitors were internalized into glioma tissues *in vivo* and BIP-MPC-NP had a more effective BBB permeability than BIP-NP.

Methods (revised part):

The tissue slice distribution

After injection of cyanine 5.5 (Cy5.5)-labeled BIP-MPC-NP or Cy5.5-labeled BIP-NP for 5 hours, the tumor-bearing brains were removed and further fixed, followed by dehydration in 15% sucrose and 30% sucrose at 4°C. Frozen coronal sections of 20 μ m in thickness were prepared and processed for fluorescence imaging.

8. Why was the temozolomide intraperitoneally injected? p.o. or i.v. is more suggestive to mimic the clinical treatment.

Response: Thank you for your valuable comments. In clinical treatments, temozolomide had several adverse events including vomiting, allergic reaction and others. When patients were treated with temozolomide by oral administration, they have to take ondansetron (32 mg) to prevent nausea and vomiting³. Allergic reaction, pain, redness, swelling at the injection site may occur while administering temozolomide intravenously. These adverse events were difficult to be treated in animal experiments. Therefore, most researchers preferred intraperitoneal injection as an alternative route of temozolomide administration in experiments on nude mice^{4,6}. We had followed the procedure of temozolomide administration from their published works.

9. Which kind of cells was used for *in vivo* treatment study?

Response: Thank you for your valuable comments. The LN229R cells were used for *in vivo* treatment study. In the original manuscript, we have described that we transplanted LN229R cells into BALB/c nude mice in the **Results** section. In the revised manuscript, we also revised the **Methods** section in more details to clarify the presentation.

Methods (revised part):

***In vivo* tumor inhibition**

Four-week-old female athymic BALB/c nude mice were purchased from Beijing Vital River Laboratory Animal Technology Co., Ltd. (Beijing, China) and were randomly divided into five groups (ten mice per group). A total of 3×10^5 LN229R cells per mouse were stereotactically injected into the brain. After 3 days of the surgery, the mice were treated with TMZ (60 mg/kg/day) by intraperitoneal injection every day. EBP-MPC-NP (4 μ M), MBP-MPC-NP (4 μ M) or BIP-MPC-NP (4 μ M) was injected intravenously every other day after the surgery. The intracranial tumors were measured with Magnetic Resonance Imaging 9.4 T horizontal Bruker magnets (Bruker Corporation, Beijing, China) and the consecutive sections (0.7 mm) were obtained at the seventh, fifteenth and twenty-first day after tumor implantation. The craniocaudal diameter (dcc), the anteroposterior diameter (dap) and the largest lateral diameter (dl) were measured by MRI. Diameter-based measurements were computed ($V = dcc \times dap \times dl \times \pi / 6$) to calculate the volume (V). The brain tissues and other major organs were embedded in paraffin and sectioned at a thickness of 2 μ m for immunohistochemistry (IHC) assay and hematoxylin-eosin staining (H&E staining). The blood samples were collected at the same time for serum enzymes assays at the end of different nanoinhibitor treatments. All procedures were approved by the Committee on the Ethics of Animal Experiments of Harbin Medical University.

10. The *in vivo* distribution of the nanoparticles should be performed in glioma bearing mice.

Response: Thank you for your suggestions in helping us to improve our work. We performed the fluorescence assay to compare the signal of Cy5.5-labeled BIP-MPC-NP and the signal of Cy5.5-labeled BIP-NP (without MPC) in glioma bearing mice brains. The xenograft gliomas were detected by luminescence and the *in vivo* distribution of Cy5.5-labeled BIP-MPC-NP and Cy5.5-labeled BIP-NP was detected by fluorescence. These data were showed in revised **Figure 2f-h**. We also revised the description in the **Results** section.

Revised Figure 2f-h:

Results (revised part):

To further evaluate the capability of the nanoinhibitors in crossing BBB *in vivo*, we employed LN229R tumor-bearing mice to perform this study. Cy5.5-labeled BIP-MPC-NP and BIP-NP were injected via the tail vein at a dosage of 100 μ L (4 μ M). The mice were then sacrificed, and the brains were harvested for *ex vivo* imaging 5 hours post the injection. According to the results (Figure 2f-h, Supplementary Figure 6d-e), the fluorescence signal of Cy5.5-labeled BIP-MPC-NP was observed clearly in the mouse brains and showed a 3.2-fold higher increase than BIP-NP, indicating effective BBB permeability.

11. Please add the detailed particle size and PDI of BIP-MPC-NP and MPC-NP separately.

Response: Thank you for your important suggestions. In the revised manuscript, all samples were dissolved in a low ionic strength (10 mM) phosphate buffer (pH = 7.4). The PDI value was assessed according to previous works (a PDI value of 0.1 ~ 0.25 indicates a narrow size distribution, whereas a value greater than 0.5 indicates a broad distribution)⁷. As shown in revised **Supplementary Table 5**, the BIP-MPC-NP dissolved in PB had a small average diameter 23.65 (\pm 2.23) nm and a narrow PDI 0.26 (\pm 0.06) nm. We have added these data in the revised manuscript in the revised **Results** section.

Results (revised part):

The transmission electron microscope (TEM) image of the nanoinhibitors showed a spherical shape with an average diameter of 23.65 (\pm 2.23) nm, which was further confirmed with dynamic

light scattering (DLS) measurements (Figure 1c-d, Supplementary Figure 5a). The DLS and PDI values of nanoinhibitors were also showed in *Supplementary Table 5*.

Revised Supplementary Table 5:

Supplementary Table 5. The DLS and PDI values of nanoinhibitors

	MPC-NP	EBP-MPC-NP	MBP-MPC-NP	BIP-MPC-NP
DLS	18.84 ± 1.65 (nm)	24.56 ± 2.29 (nm)	22.67 ± 2.20 (nm)	23.65 ± 2.23 (nm)
PDI	0.18 ± 0.02	0.26 ± 0.03	0.20 ± 0.04	0.26 ± 0.06

12. Scale bars should be added to Figure 3d.

Response: Thank you for your valuable comments. The scale bars were added to the original Figure 3d. To make a clearer way for presenting data, we have reformatted the *Figure 1* and *Figure 2*. The original Figure 3d was moved to the revised *Figure 1j*. Thank you again for your important comments.

Revised Figure 1:

Response to Reviewer #2:

1. This manuscript by Meng et al, reports the development and implementation of dual-functionalized brain targeting nano-inhibitors in glioma models that exhibit resistance to TMZ. The authors elected to target EGFR and MET signaling pathways. The authors have provided substantial data, which are of high quality. The main concern with the overall aim of this work is that the authors have not compared the response of the TMZ resistant models vs the TMZ sensitive models. Their overall premise lies on this distinction, thus, it is critical that the authors show differential responses to the treatment utilizing these two model systems.

Response: Thank you very much for your concerns and critical comments.

As the reviewer concerned, we established the TMZ resistant GBM cells named LN229R, U87MGR and HG9R through the TMZ treatment on LN229, U87MG and patient-derived GBM cells HG9 three kinds of relatively TMZ sensitive GBM cells. The half maximal inhibitory concentration (IC50) of TMZ resistant glioma cells were higher than that of parental cells. The similar method and process has been carried out in our previous work^{8,9}.

We performed the gene expression profiling of the TMZ resistant GBM cells LN229R and the parental cells LN229 and observed that EGFR and MET expression were much higher in LN229R cells compared with those in LN229 cells. *In vivo* immunofluorescence and immunohistochemistry showed that the expression and phosphorylation levels of EGFR and MET in the glioma tissues derived LN229R cells are higher than those in LN229 cells (revised ***Supplementary Figure 2b-c***). Based on these results, we developed a novel nanoinhibitor, BIP-MPC-NP, which could simultaneously mitigate EGFR and MET activation and we used it combined with TMZ to treat the TMZ resistance GBM cells, but not the parental cells with relatively TMZ sensitivity and limited activation levels of EGFR and MET. Compared with the TMZ resistance GBM cells, the parental cells or animal models with relatively TMZ sensitivity were difficult to tolerate the treatment with the same concentration of TMZ^{6,8}. Therefore, in our present work, we only evaluated the TMZ sensitivity improvement of nanoinhibitor, BIP-MPC-NP, for TMZ resistance cells or animal models, but not relatively TMZ sensitive GBM cells or animal models.

Revised Supplementary Figure 2:

Supplementary Figure 2

2. The authors should also provide histological data in relation to the size of the tumors at the time of treatment. This is critical as it would be important to demonstrate the presence of a tumor mass at the time the treatment starts.

Response: Thank you for your valuable comments. In the original *Figure 8*, we have provided the MRI images to demonstrate the presence of a tumor mass. The first row of MRI images represented the sizes of xenograft gliomas in each group. The tumor volume data was displayed in histograms. These parts of data were moved to the revised *Figure 6*. Meanwhile, according to the comment, we provided more histological data such as H&E staining of tumor slices to evaluate the tumor size at the 21st day after the treatment starts in the revised *Figure 6a*.

Revised Figure 6:

3. The authors also need to provide neuropathological data from the brain of the animals that were treated and died of tumor burden, and compare the neuropathology with control, non-treated animals.

Response: Thank you for your valuable comments. In our work, we observed that the TMZ+BIP group showed significantly reduced expression levels of p-EGFR and p-MET in glioma tissues by immunofluorescence and immunohistochemistry assays (revised *Supplementary Figure 17*) and that the higher expression of TTP and γ HAX, and the lower expression of E2F1, CHK1, CHK2,

p53, RAD50 and RAD51 in glioma tissues were also observed in TMZ+BIP group compared with those in other groups. And we have also quantified these results (revised *Supplementary Figure 18*, revised *Supplementary Figure 19*).

According to the comments, we provided more neuropathological data such as H&E staining in the revised *Figure 6a*.

Revised Supplementary Figure 17:

Revised Supplementary Figure 18:

Revised Supplementary Figure 19:

4. An analysis of immune cellular infiltrated by immunohistochemistry should also be presented.

Response: Thank you very much for your comments. Since our nanoinhibitor did not show significant immune response after repeated administration in tumor-bearing mice (revised *Supplementary Table 6*), we considered that it might not be the point to analyze the immune cellular by immunohistochemistry. In our previous work, we have systematically studied the immune cellular of glioma microenvironment⁹. Normally, TMZ-resistant gliomas are correlated with immunosuppression, response to cytokines and other immune cellular processes. These gliomas appear to avoid host immune-mediated elimination through activation of immunosuppression. The converting immune microenvironment releases a plethora of cytokines and chemokines to communicate with other cells and thereby to orchestrate immune responses the relative abundance of individual immune cell types differed between DNA damage repair alterations. However, in order to estimate the functions of nanoinhibitors on human malignant gliomas, we transplanted human glioma cells into BALB/c nude mice which exhibit immunodeficient phenotype. Since BALB/c nude mice are lack of immune cellular, xenograft gliomas could be established without immune attack from different species.

In this manuscript, we assessed the immune response by hematology assay to demonstrate the safety of our nanoinhibitors.

Revised Supplementary Table 6:

Supplementary Table 6. Blood routine analysis in blood samples of mice bearing orthotopic glioma at the end of experiment.

Group	WBC (K/ μ L)	RBC (M/ μ L)	Hbg (g/L)	Plt ($\times 10^9$ /L)
Control	8.41 \pm 0.11	10.21 \pm 0.33	146.31 \pm 2.21	1060.98 \pm 33.29
TMZ	8.87 \pm 0.29	10.90 \pm 0.21	141.01 \pm 0.30	982.52 \pm 66.61
TMZ+EBP	9.08 \pm 0.21	9.93 \pm 0.26	151.22 \pm 1.29	990.35 \pm 56.98
TMZ+MBP	7.95 \pm 0.21	9.92 \pm 0.28	140 \pm 0.91	1103.25 \pm 59.01
TMZ+BIP	8.93 \pm 0.16	10.18 \pm 0.29	143.12 \pm 1.95	977.69 \pm 51.40

5. DNA damage should be demonstrated by the presence of γ H2AX foci *in vivo*, in response to the treatment and the foci should be quantified.

Response: Thank you for your valuable comments. The markers for assessing DNA damage repair are multiple such as Rad50, p53, γ H2AX. In the original manuscript, we had chosen comet assay to evaluate the DNA damage *in vitro* (data were shown in the original Figure 5c-d). We also evaluated the expression of DNA damage repair molecules including Chk1, Chk2, Rad50, Rad51

and p53 to reflect the DNA damage *in vivo* (data were shown in original *Supplementary Figure 7*, *Supplementary Figure 8*, *Supplementary Figure 9*). In the revised manuscript, these parts of data were moved to the revised *Figure 3e*, *Supplementary Figure 8*, *Supplementary Figure 9*, *Supplementary Figure 10* and *Supplementary Figure 11*.

According to the valuable comment, we added estimation of the expression of γ H2AX *in vitro* by Western blot (revised *Figure 3d*) and the presence of γ H2AX foci *in vivo* by immunohistochemistry on xenograft glioma slices and quantified the results in the revised *Supplementary Figure 19*.

Revised Figure 3d:

Revised Supplementary Figure 19:

6. In addition, this paper is very difficult to read, the figure legends are poorly described and lack details required to follow content. The paper would need to be thoroughly re-written to clarify the way the data is presented and also provide details related to the figures presented.

Response: Thank you for your critical comments. We have refined the manuscript language and revised the understanding description. More details were added into both figure legends and other sections to improve the manuscript. The results were reordered make this manuscript easier to read. We also send this manuscript for language editing services of Springer Nature Author Services. We sincerely apologize for the inconvenience in reading our manuscript and we hope that the language would not affect the quality and the assessment of our work. We also believed that with these improvements, this manuscript will be more concise and clearer when presenting data. Thank you again for the patience and effort on our manuscript.

References

1. Gaillard, P. J.; de Boer, A. G., Relationship between permeability status of the blood-brain barrier and in vitro permeability coefficient of a drug. *Eur J Pharm Sci* **2000**, *12* (2), 95-102.
2. Han, L.; Liu, C.; Qi, H.; Zhou, J.; Wen, J.; Wu, D.; Xu, D.; Qin, M.; Ren, J.; Wang, Q.; Long, L.; Liu, Y.; Chen, I.; Yuan, X.; Lu, Y.; Kang, C., Systemic Delivery of Monoclonal Antibodies to the Central Nervous System for Brain Tumor Therapy. *Adv Mater* **2019**, e1805697.
3. Baker, S. D.; Wirth, M.; Statkevich, P.; Reidenberg, P.; Alton, K.; Sartorius, S. E.; Dugan, M.; Cutler, D.; Batra, V.; Grochow, L. B.; Donehower, R. C.; Rowinsky, E. K., Absorption, metabolism, and excretion of ¹⁴C-temozolomide following oral administration to patients with advanced cancer. *Clin Cancer Res* **1999**, *5* (2), 309-17.
4. Tentori, L.; Leonetti, C.; Scarsella, M.; d'Amati, G.; Portarena, I.; Zupi, G.; Bonmassar, E.; Graziani, G., Combined treatment with temozolomide and poly(ADP-ribose) polymerase inhibitor enhances survival of mice bearing hematologic malignancy at the central nervous system site. *Blood* **2002**, *99* (6), 2241-4.
5. Kato, Y.; Holm, D. A.; Okollie, B.; Artemov, D., Noninvasive detection of temozolomide

in brain tumor xenografts by magnetic resonance spectroscopy. *Neuro Oncol* **2010**, *12* (1), 71-9.

6. Han, B.; Cai, J.; Gao, W.; Meng, X.; Gao, F.; Wu, P.; Duan, C.; Wang, R.; Dinislam, M.; Lin, L.; Kang, C.; Jiang, C., Loss of ATRX suppresses ATM dependent DNA damage repair by modulating H3K9me3 to enhance temozolomide sensitivity in glioma. *Cancer Lett* **2018**, *419*, 280-290.

7. Patravale, V. B.; Date, A. A.; Kulkarni, R. M., Nanosuspensions: a promising drug delivery strategy. *J Pharm Pharmacol* **2004**, *56* (7), 827-40.

8. Wu, P.; Cai, J.; Chen, Q.; Han, B.; Meng, X.; Li, Y.; Li, Z.; Wang, R.; Lin, L.; Duan, C.; Kang, C.; Jiang, C., Lnc-TALC promotes O(6)-methylguanine-DNA methyltransferase expression via regulating the c-Met pathway by competitively binding with miR-20b-3p. *Nature communications* **2019**, *10* (1), 2045.

9. Meng, X.; Duan, C.; Pang, H.; Chen, Q.; Han, B.; Zha, C.; Dinislam, M.; Wu, P.; Li, Z.; Zhao, S.; Wang, R.; Lin, L.; Jiang, C.; Cai, J., DNA damage repair alterations modulate M2 polarization of microglia to remodel the tumor microenvironment via the p53-mediated MDK expression in glioma. *EBioMedicine* **2019**, *41*, 185-199.

REVIEWERS' COMMENTS:

Reviewer #1 (Remarks to the Author):

The comments are well addressed. It can be accepted without further modification.

Reviewer #2 (Remarks to the Author):

This revised manuscript by Meng et al., has been extensively revised, after substantial additional experimentation. The authors have addressed all the previous comments, added further experimentation and reported novel results to address the concerns raised during the previous round of reviews. This is now a very strong manuscript, which reports high quality and novel data. Further, their conclusions are in line with the results presented. The work reported is of high interest to the readership of Nature Communications.

Dr Maria G. Castro

Point by point response to reviewers' comments:

Reviewer #1 (Remarks to the Author):

The comments are well addressed. It can be accepted without further modification.

Response: Thank you very much for your valuable comments and your patience during revision.

Reviewer #2 (Remarks to the Author):

This revised manuscript by Meng et al., has been extensively revised, after substantial additional experimentation. The authors have addressed all the previous comments, added further experimentation and reported novel results to address the concerns raised during the previous round of reviews. This is now a very strong manuscript, which reports high quality and novel data. Further, their conclusions are in line with the results presented. The work reported is of high interest to the readership of Nature Communications.

Response: Thank you very much for your valuable comments in improving our work.